# Sis1 potentiates the stress response to protein aggregation and elevated temperature

Courtney L. Klaips[1,2], Michael H. M. Gropp [1], Mark S. Hipp [2,3] & F. Ulrich Hartl [1✉]

Cells adapt to conditions that compromise protein conformational stability by activating various stress response pathways, but the mechanisms used in sensing misfolded proteins remain unclear. Moreover, aggregates of disease proteins often fail to induce a productive stress response. Here, using a yeast model of polyQ protein aggregation, we identified Sis1, an essential Hsp40 co-chaperone of Hsp70, as a critical sensor of proteotoxic stress. At elevated levels, Sis1 prevented the formation of dense polyQ inclusions and directed soluble polyQ oligomers towards the formation of permeable condensates. Hsp70 accumulated in a liquid-like state within this polyQ meshwork, resulting in a potent activation of the HSF1 dependent stress response. Sis1, and the homologous DnaJB6 in mammalian cells, also regulated the magnitude of the cellular heat stress response, suggesting a general role in sensing protein misfolding. Sis1/DnaJB6 functions as a limiting regulator to enable a dynamic stress response and avoid hypersensitivity to environmental changes.

[1] Department of Cellular Biochemistry, Max Planck Institute of Biochemistry, Am Klopferspitz 18, 82152 Martinsried, Germany. [2] Department of Biomedical Sciences of Cells and Systems, University Medical Center Groningen, University of Groningen, Antonius Deusinglaan 1, 9713 AV Groningen, The Netherlands. [3] School of Medicine and Health Sciences, Carl von Ossietzky University Oldenburg, Oldenburg, Germany. ✉email: uhartl@biochem.mpg.de

Various adaptive stress response pathways operate in the cytosol and within subcellular compartments to maintain protein homeostasis (proteostasis) in a wide range of metabolic and environmental conditions[1–5]. During conformational stress, such as exposure to increased temperature, the pressure on the cellular proteostasis system increases due to an enhanced propensity of newly-synthesized and preexistent proteins to misfold and aggregate, forming potentially toxic species. The cytosolic heat stress response (HSR) allows cells to adapt to high temperature by increasing the capacity of the proteostasis network via transcription of quality control (QC) components[6], prominently including stress proteins that function as molecular chaperones[7,8].

The HSR is controlled by HSF1 as the master transcription factor, which is maintained in an inactive state under non-stress conditions, primarily by binding to chaperones such as Hsp70[8–11]. According to current models, during stress, these chaperone repressors are titrated away from HSF1 by aberrant protein species, thus freeing HSF1 to execute its transcriptional program[12–14]. Once sufficiently augmented, chaperones eventually re-bind to HSF1, thus creating a feedback loop for recovery[8,12,15,16]. While the initial activation of the HSR is critical for coping with stress conditions, its shut-off appears to be equally important, as chronic activation of the HSR leads to maladaptation, with an inability to cope with additional stressors and disturbance of biological development in higher organisms[17–19].

Despite previous studies on the kinetics of the HSR[16,20], the molecular mechanisms adjusting the magnitude and the speed of recovery of the stress response are not well understood. The exact nature of the aberrant protein species critical for HSF1 activation by chaperone titration has been questioned. For example, it remains unclear whether stress-induced aggregates can be sensed directly, or whether other changes to the cellular environment are required, such as a lowering of cytosolic pH[21].

Numerous neurodegenerative conditions, including Huntington's and Parkinson's disease, are associated with the buildup of amyloid-like aggregates, which are highly ordered, cross-β-sheet structures[22]. Their accumulation is typically independent of acute stress conditions but may be facilitated by an age-dependent decline in cellular proteostasis capacity[2,23,24]. These pathological aggregates exert multiple toxic effects, including aberrant interactions with endogenous proteins that lead to sequestration of key cellular factors, including chaperones[25–32]. Remarkably, disease related aggregates often fail to trigger a beneficial stress response[25,33,34], although aggregation and toxicity can be suppressed through upregulation of individual chaperone components or HSR activation by chemical compounds[35–42]. This raises the general question as to the mechanisms used by cells in sensing conformational stress.

Identifying factors that enable cells to induce a potent stress response to amyloid-like aggregation may provide insight into the general mechanism of stress regulation. We expressed polyglutamine (polyQ)-expanded Huntingtin exon-1 as a model disease protein in yeast cells, where its aggregation can be observed without overt toxicity[26,37,43,44]. While the polyQ aggregates triggered only a weak HSR, expression of members of a chaperone library identified the Hsp40, Sis1, as a limiting factor required for cells to mount a robust, aggregation-specific stress response. Sis1, an essential Hsp70 co-chaperone that shuttles between cytosol and nucleus[26], prevented the formation of dense inclusions and directed soluble oligomers of polyQ into cloud-like condensates. Hsp70 accumulated within these condensates in a liquid-like state, resulting in HSR activation. Sis1, and the homologous DnaJB6 in mammalian cells, also regulated the magnitude of the HSR to elevated temperature, with normally limited Sis1/DnaJB6 levels acting as a key regulatory element in avoiding hypersensitivity to environmental changes.

## Results

**Aggregation of polyQ protein does not trigger a stress response.** To study the activation of the cytosolic stress response by protein aggregates, we employed an established yeast model of polyQ length-dependent protein aggregation[26,37,43,44]. We expressed huntingtin (Htt) exon-1 fragments with normal (20Q) and expanded (97Q) polyQ tracts as fusion proteins with mCherry and an N-terminal Myc-tag under control of the *GAL1* promoter (Fig. 1a). Expression of Htt20Q resulted in diffusely distributed protein, whereas Htt97Q formed large inclusions that were localized primarily in the cytosol[26,43], and did not co-localize with a nuclear targeted GFP protein (Fig. 1b). A fraction of Htt97Q formed SDS insoluble aggregates, as detected in cell lysates by filter retardation assay[45] (Supplementary Fig. 1a), consistent with the presence of amyloid-like fibrils and densely packed amorphous aggregates in the inclusions[46,47]. As reported previously, polyQ aggregation in this system was not accompanied by a major growth impairment[26,43,46] (Supplementary Fig. 1b). A pronounced polyQ length-dependent growth defect in yeast is only observed upon expression of Htt exon-1 lacking the poly proline (PP) region C-terminal to the polyQ tract[43,48].

We next determined whether the formation of polyQ aggregates triggered a cytosolic stress response. We utilized a LacZ-based reporter under control of a minimal promoter containing a heat shock element (HSE) from the Hsp70 *SSA3* gene ($P_{HSE}$LacZ)[49]. Induction of this reporter was proportional to the magnitude of the temperature stress (Fig. 1c). It was specific to the HSR pathway, as overexpression of HSF1 alone induced activity, but not treatment with DTT, a known inducer of the unfolded protein response (UPR) of the ER (Fig. 1c). Expression of either Htt20Q or Htt97Q at normal growth temperature (30 °C) did not trigger a robust stress response (Fig. 1d). Although expression of Htt97Q caused a slight HSR induction, this response was not comparable in magnitude to that induced by heat stress (Fig. 1c, d), consistent with poor HSR induction by polyQ aggregates in other model systems[33,34,50]. We ruled out effects of polyQ expression on synthesis or folding of the LacZ reporter, as an identical reporter under control of a galactose-inducible promoter did not show a significant polyQ length-dependent reduction of activity (Supplementary Fig. 1c).

The failure of Htt97Q aggregates to induce a robust HSR reflected either an inability of the cells to recognize the aberrant protein, or an inhibition of HSF1 signaling. To distinguish these possibilities, we exposed Htt97Q expressing cells to a mild heat stress at 37 °C for 1 h. The magnitude of the stress response in presence of Htt97Q was comparable to that in control cells or cells expressing soluble Htt20Q, based upon both reporter induction (Fig. 1d, red bars) and the levels of stress inducible chaperones (Supplementary Fig. 1d), indicating that polyQ expressing cells could still sense and respond to heat stress. Indeed, HSF1 overproduction was sufficient to cause induction of the LacZ reporter in both Htt20Q and Htt97Q cells (Fig. 1d, black bars). This response was biologically functional, as it allowed cells to survive a lethal heat treatment at 50 °C (Supplementary Fig. 1e, right). Taken together, these results suggest that ineffective HSR induction by Htt97Q is due to the inability of the proteostasis network to sense the polyQ aggregates, not to an overall inhibition of the HSR pathway.

**Sis1 enables stress response activation by expanded polyQ.** Overexpression of individual chaperones can modulate polyQ aggregation and fibril formation, and mitigate toxic effects[15,37,40].

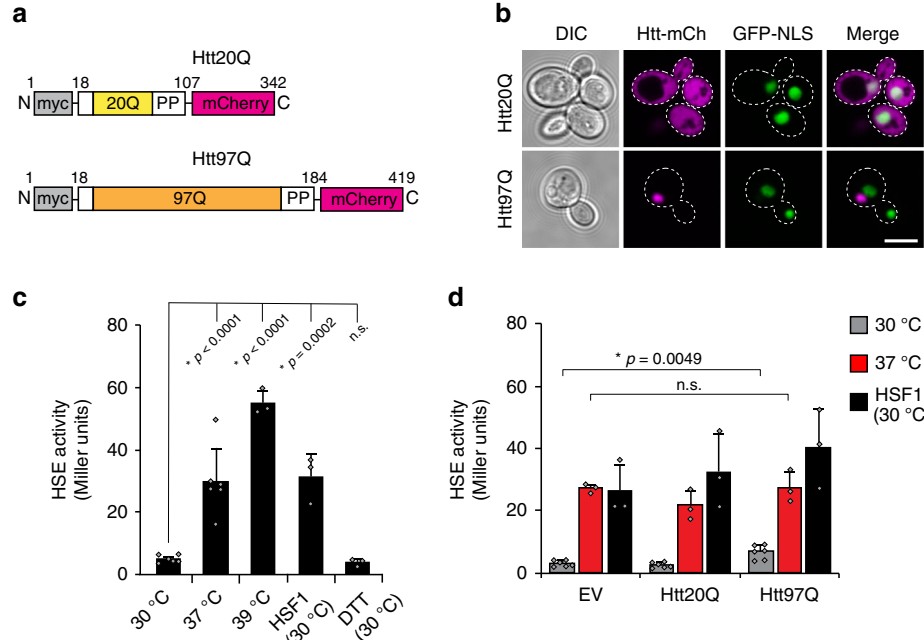

**Fig. 1 Aggregation of polyQ expansion protein does not trigger a stress response in yeast. a** Schematic of constructs expressing Huntingtin (Htt) exon-1 with either 20 Q (Htt20Q) or 97 Q (Htt97Q), tagged with a N-terminal Myc epitope and fused to mCherry under a galactose-inducible promoter. PP, poly-proline. **b** Localization of Htt constructs in cells. Confocal images of cells containing Htt20Q or Htt97Q constructs and a nuclear targeted GFP after growth in inducing media for ~21 h. Scale bar = 5 μm. Experiments were performed in triplicate, representative images shown. **c** Response of the LacZ reporter to elevated temperature. β-galactosidase (β-Gal) activity was measured in cells expressing a LacZ reporter under the control of a minimal promoter containing a heat shock element (HSE) from *SSA3* ($P_{HSE}$LacZ) grown at 30 °C for ~20 h, followed by a shift to 37 °C or 39 °C or treatment with DTT (2 mM) for 1 h. As an additional control, β-Gal activity was measured in cells co-expressing HSF1 and grown at 30 °C. Here and throughout, LacZ activity is reported in standard Miller Units (see "Methods" section for details). Data represent mean + SD from three independent experiments. *p values were calculated using Dunnett's multiple comparisons *t*-test to 30 °C. n.s. not statistically significant. **d** The heat-induced stress response (HSR) remains active in cells expressing Htt constructs. β-Gal activity was measured in cells expressing EV, Htt20Q, or Htt97Q at 30 °C as in **c** either maintained at 30 °C or shifted to 37 °C for 1 h, or co-transformed with HSF1. Data represent mean + SD from three independent experiments. *p values are reported for unpaired, two-sided *t*-test. n.s. not statistically significant.

We considered the possibility that augmentation of specific QC components might be required for cells to efficiently recognize, and ultimately respond to, the presence of these aggregates. To test this hypothesis, we conducted a systematic screen for cha-perone factors that when expressed at elevated levels allow for stress response induction by Htt97Q. Of the ~67 chaperones in yeast, 50 were included in our 2 μ library screen (Supplementary Table 1). We initially identified six factors as allowing for stress response induction upon expression with Htt97Q (Fig. 2a, Sup-plementary Fig. 2a and Supplementary Table 1). A counter screen with cells expressing Htt20Q revealed that only expression of the Hsp40, Sis1, resulted in a significant stress response induction that was specific to Htt97Q (Supplementary Fig. 2a and Supple-mentary Table 1). Co-expression of 2 μ plasmids has been reported to lead to variations in copy number of polyQ expanded huntingtin[51]. We therefore validated the effects of the primary screen using a centromeric expression vector for Sis1 under the strong *GPD* promoter (Fig. 2b). Sis1 overexpression by ~5-fold allowed for a robust Htt97Q-specific stress response (Fig. 2b and Supplementary Fig. 2b), while overexpression of the cytosolic Hsp40 homolog, Ydj1, induced only a moderate stress response (Fig. 2b), despite being ~5-fold overexpressed, similar to Sis1 (Supplementary Fig. 2b). Note that increased LacZ reporter activity was not due to enhanced folding of β-galactosidase mediated by excess Sis1 (Supplementary Fig. 2c).

Interestingly, Sis1 overexpression has been shown to ameliorate the toxic effects of polyQ-expanded Htt exon-1 lacking the PP region[48,52]. Indeed, while there were higher basal levels of HSR

induction in cells expressing Htt97QΔP compared to Htt97Q, overexpression of Sis1 markedly enhanced this stress response (Supplementary Fig. 2d), suggesting that the beneficial effects of Sis1 are due (at least in part) to HSR activation.

Sis1 has also been reported to play a role in maintaining endogenous yeast prions, such as the [*PIN*+] prion conferred by the Rnq1 protein[53,54], and Htt aggregation in yeast is known to be dependent on the presence of Rnq1 aggregates[55]. We therefore tested whether Sis1 expression altered the prion status of Htt97Q expressing cells. We transiently expressed fluorescently labeled Rnq1 to probe the prion status of our strains with and without Sis1 overexpression. In both cases, Rnq1 aggregation could be observed, as opposed to [*pin*−] control cells, suggesting that Sis1 overexpression was not simply curing the prion in our conditions (Supplementary Fig. 2e).

Hsp40s generally act as co-chaperones by recruiting Hsp70 to specific substrates[56–58]. Yeast have two major Hsp40s in the cytosol, Ydj1 and Sis1, that share an Hsp70 interacting J-domain but otherwise vary in domain composition, substrate specificity and cellular abundance (Supplementary Fig. 2f)[59]: Ydj1 is a type I Hsp40, defined as containing a zinc-finger like region (ZFLR) in its C-terminal domain, whereas Sis1 is a type II Hsp40, lacking this region. Mammals have several type II Hsp40 homologs that we hypothesized could substitute for Sis1 in the observed stress response induction. We focused on DnaJB1 and DnaJB6 (Supplementary Fig. 2f). In terms of domain structure, DnaJB1 is more homologous to Sis1, as both contain a longer C-terminal domain, lack a ZFLR, and contain a dimerization domain.

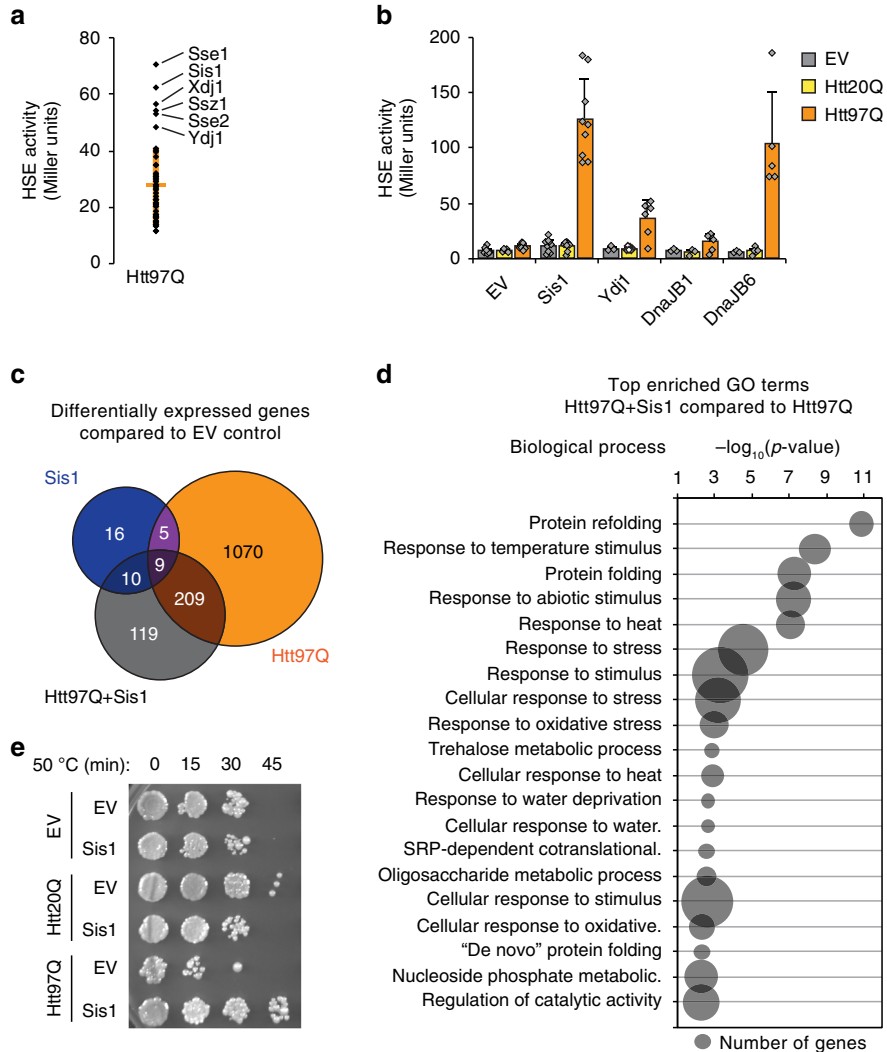

**Fig. 2 Sis1 enables stress response activation by polyQ expansion protein. a** Screen for factors allowing induction of the HSR by Htt97Q. β-Gal activity is shown for P$_{HSE}$LacZ containing cells expressing Htt97Q and overexpressing one of 50 yeast chaperones. Orange line represents screen average, orange bars represent SD. Significant hits falling outside the SD are identified. See also Supplementary Table 1. **b** Sis1 overexpression enables stress response activation. β-Gal activities were measured in P$_{HSE}$LacZ reporter cells expressing empty vector (EV), Htt20Q or Htt97Q (P$_{GAL}$) and overexpressing the Hsp40 chaperone proteins indicated under the *GPD* promoter. Data represent mean + SD from at least three independent experiments. **c** Transcriptional response to Htt97Q. The numbers of genes differentially expressed compared to control cells are represented in a Venn diagram after transcriptome analysis of control cells, cells overexpressing Sis1, and cells expressing Htt97Q with and without overexpression of Sis1. EV empty vector. Experiments were performed in triplicate. See also Supplementary Table 2. **d** Genes upregulated with Sis1 overexpression in Htt97Q containing cells correspond to HSR associated pathways. The top enriched GO biological processes (*p* < 0.005) for differentially expressed genes upregulated in Htt97Q cells upon Sis1 overexpression. *p* values were calculated using the GOseq R package, which corrects for gene length bias. See also Supplementary Table 3. **e** Sis1 mediated HSR induction by Htt97Q has protective effects. Cells were grown at 30 °C after expression of Htt20Q, Htt97Q, or EV control with or without Sis1 overexpression and exposure to heat shock at 50 °C for the indicated times. The experiment was performed in triplicate; a representative result is shown.

DnaJB6, however, like Sis1, has been shown to interfere with polyQ aggregation and toxicity in various systems[36,60–62]. Strikingly, overexpression of DnaJB6 enabled yeast cells to activate the stress response to Htt97Q (Fig. 2b). Note that DnaJB6 exists primarily as two splice variants, a longer, exclusively nuclear form (DnaJB6a) and the shorter, primarily cytosolic form (DnaJB6b) used here[63]. The ability of DnaJB1 to substitute for Sis1 in stress response induction was less clear due to a lower efficiency in DnaJB1 overexpression (Supplementary Fig. 2g).

To assess the cellular response to Htt97Q upon Sis1 overexpression, we performed a transcriptome analysis. Overexpression of Sis1 alone resulted in the differential regulation of relatively few genes with no specific pathway enrichment (Fig. 2c

and Supplementary Table 2). As expected, expression of Htt97Q alone led to widespread transcriptional changes[64] (Fig. 2c), including upregulation of multiple genes involved in nuclear processes, such as ribosome biogenesis[28,65] (Supplementary Table 2). Upon co-overexpression of Sis1, the total number of differentially regulated genes decreased (Fig. 2c), and evidence of nuclear process dysregulation was lost (Supplementary Table 2). Instead, many processes associated with the HSR were upregulated (Supplementary Table 2). To explore these changes in more detail, we further analyzed cells expressing Htt97Q alone compared to cells expressing Htt97Q with Sis1 overexpression. Many examples of HSF1 regulated factors (40 of 67 HSF1 regulated genes)[66] as well as factors of the MSN2/4 arm of the stress response (52 of 213 MSN2/4 regulated genes)[67] were

significantly upregulated with Sis1 expression (Supplementary Fig. 2h). Of the 23 genes that were upregulated more than 2-fold upon co-expression of Htt97Q and Sis1 compared to Htt97Q alone (Supplementary Table 3), 12 belong to major chaperone/co-chaperone families (Supplementary Fig. 2i)[68], with additional genes belonging to minor cofactor classes. Among these genes, there is considerable connectivity (Supplementary Fig. 2i), suggesting that specific pathways are upregulated. Consistently, GO term analysis confirmed the top hits as processes associated with heat shock (Fig. 2d and Supplementary Table 3). Importantly, this stress response was also biologically effective. Although Htt97Q cells were more heat sensitive than Htt20Q cells, the stress response observed upon combined expression of Sis1 and Htt97Q protected cells against a lethal heat stress exposure (Fig. 2e). In contrast, expression of Sis1 alone or with Htt20Q was not protective (Fig. 2e).

PolyQ aggregation in the cytosol may affect proteostasis in other compartments, such as the ER[69,70]. The mRNA sequencing results suggested that the effect of Sis1 is specific to the cytosolic stress response (Supplementary Fig. 2h and Supplementary Table 3)[71]. Indeed, Sis1 overexpression did not activate a LacZ reporter with a UPR[ER] stress-inducible promoter[72], nor did it sensitize cells for UPR[ER] induction by Htt97Q (Supplementary Fig. 2j). In contrast, treatment with the ER stressor DTT caused robust induction of the UPR[ER] reporter (Supplementary Fig. 2j).

**Sis1 induces cloud-like polyQ condensates**. To understand how Sis1 enables induction of the stress response pathway by polyQ aggregates, we analyzed whether Sis1 modulates the morphology of the aggregates. While the majority of control cells formed one or several sharply delineated cytosolic inclusions, Sis1 overexpression typically resulted in the formation of a single large and diffuse appearing cytosolic Htt97Q cluster per cell (Fig. 3a). Density measurements revealed that, based on mCherry fluorescence, the concentration of Htt97Q inside these cloud-like condensates was ~4-fold lower than in the inclusions of control cells (Fig. 3b). Since comparisons were made between cells expressing similar total levels of Htt97Q, this suggests that Sis1 mediates the formation of a less dense aggregate state. Sis1 has previously been reported to shuttle between the cytosol and nucleus[26,73]. Nuclear Sis1 could be observed in Htt20Q expressing cells using a strain in which the endogenous *SIS1* was replaced by *SIS1-GFP* (Supplementary Fig. 3a). In contrast, Sis1 was recruited to the cytosolic polyQ inclusions in Htt97Q expressing cells, as previously reported[26], and also accumulated in the Htt97Q condensates upon Sis1 overexpression (Fig. 3c).

Whereas in control cells ~50% of Htt97Q was insoluble upon cell fractionation (in the absence of SDS), almost all polyQ protein was recovered in the soluble fraction upon Sis1 overexpression, indicating that the condensates readily dissolved upon cell lysis (Fig. 3d). Despite this increase in solubility, fluorescence recovery after photobleaching (FRAP) experiments showed that both Htt97Q and Sis1 inside the condensates were largely immobile, displaying only a slightly higher mobility than in the polyQ inclusions of control cells (Fig. 3e). Conversely, Htt20Q recovered from photobleaching almost instantly, as expected for a mobile, soluble protein, which would move back into the bleached area within the first imaging time frame[74] (Supplementary Fig. 3b). Thus, the condensates formed with Sis1 are not liquid-like but rather behave like an immobile, yet dissociable, mesh, consistent with their irregular shape (Fig. 3a, c). Overexpression of DnaJB6 reproduced the effect of Sis1 in mediating formation of polyQ condensates, while DnaJB1, expressed to a lower level than DnaJB6, did not (Supplementary Fig. 3c).

Given the recovery of Htt97Q in the soluble fraction in Sis1 overexpressing cells, we asked whether accumulation of soluble Htt97Q is important for stress response induction. The aggregation of polyQ in yeast is dependent on the presence of endogenous prion aggregates[55] and thus expression of polyQ in [pin⁻] strains did not lead to the formation of visible polyQ foci, with or without Sis1 overexpression (Supplementary Fig. 3d). Expression of Htt97Q in [pin⁻] cells did not induce a stress response (Supplementary Fig. 3e), indicating that the presence of soluble Htt97Q alone is not sufficient for HSR activation. Furthermore, overexpression of Sis1 no longer allowed for HSR activation by Htt97Q in [pin⁻] cells (Supplementary Fig. 3e). To modulate the amount of soluble polyQ protein without changing the prion status of the cells, we expressed a Htt exon-1 construct with 48Q, which is an expanded polyQ tract below the aggregation threshold in the yeast system[37] (Supplementary Fig. 3f). Htt48Q did not induce the stress response regardless of Sis1 co-expression (Supplementary Fig. 3g). Thus, the function of Sis1 in activating the stress response requires polyQ-length dependent conformational changes that correlate with aggregate formation.

Next, we investigated whether Sis1 can mediate HSR induction in cells containing preexisting polyQ aggregates. We first allowed cells to form Htt97Q inclusions at normal Sis1 levels for ~18 h, followed by ~3 h with or without additional expression of Sis1 using a tetracycline controllable promoter (Fig. 4a). Although less pronounced, due to the shorter duration of Sis1 induction (~3 h instead of ~21 h) to avoid dilution effects by cell division, Sis1 overexpression by ~1.7 fold (Supplementary Fig. 4a) resulted in a significant stress response induction (Fig. 4b). No such induction by Sis1 was seen in control cells expressing Htt20Q (Supplementary Fig. 4b). This transient overexpression of Sis1 did not completely remodel the preexisting polyQ inclusions (Fig. 4c), which were similar in density to those observed without Sis1 overexpression (Supplementary Fig. 4c), indicating that conversion of preexisting inclusions into cloud-like condensates is not a requirement for stress response induction. Note, however, that with short-term Sis1 expression the inclusions acquired a fuzzy edge (Fig. 4c), suggesting the beginning of condensate formation around preexisting inclusions.

To distinguish between preexistent aggregates and newly-synthesized polyQ protein as the stress inducing agent, we blocked polyQ transcription by addition of glucose at the time of Sis1 induction (Fig. 4a). HSR activation was not observed under these conditions (Fig. 4b), despite the fact that polyQ inclusions persisted (Fig. 4c). Taken together, these results demonstrate that induction of the stress response requires ongoing polyQ synthesis.

To identify the polyQ species Sis1 acts on, we analyzed cell lysates by semi-denaturing agarose electrophoresis (SDD-AGE), an established method for the detection of oligomeric aggregates[75]. Htt97Q oligomers of a similar size range were detected both without and with Sis1 overexpression (Fig. 4d). Moreover, these species were found in the soluble lysate fraction (Fig. 4d), distinguishing them from the large polyQ aggregates detected by fractionation in cells with normal Sis1 levels (Fig. 3d). Sis1 overexpression resulted in a pronounced increase of Htt97Q oligomers (Fig. 4d), but was without effect on Htt20Q (Supplementary Fig. 4d). Notably, some Sis1 co-migrated with these Htt97Q oligomers, despite the presence of SDS, as visualized after long exposure of the immunoblot (Fig. 4d). In contrast, there was very little Sis1 association with polyQ oligomers at normal Sis1 levels. No high molecular weight Sis1 species could be observed in the absence of oligomeric polyQ (Supplementary Fig. 4d). We also observed Hsp40-associated polyQ oligomers upon overexpression of DnaJB6, but not of Ydj1 or DnaJB1 (Supplementary Fig. 4e, f). Together these data suggest

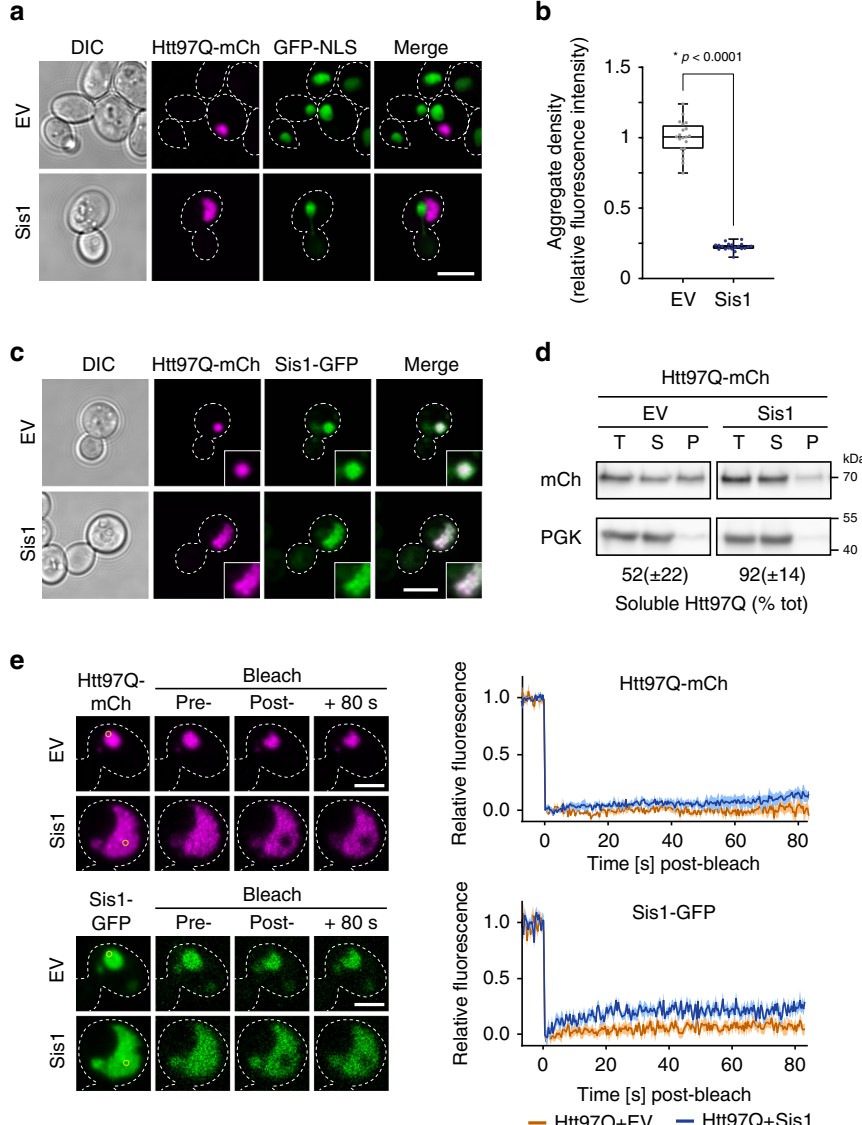

**Fig. 3 Sis1 affects physico-chemical aggregate properties. a** Htt97Q aggregates are cytosolic. Fluorescence microscopy images of cells expressing Htt97Q-mCherry without or with Sis1 overexpression. Confocal microscopy was performed on cells expressing NLS-GFP as nuclear marker. Scale bar = 5 μm. Experiments were performed in triplicate, representative images shown. **b** Htt97Q aggregates in Sis1 overexpressing cells are less dense. The average fluorescence intensity of aggregates from cells without or with Sis1 overexpression was measured using confocal microscopy. Box plots represent median and 25th and 75th percentile, and whiskers minimal and maximal values. $n = 20$ cells, *$p < 0.0001$ by unpaired, two-sided $t$-test. **c** Sis1 co-localizes with Htt97Q. Confocal images of cells containing endogenously tagged Sis1-GFP and expressing Htt97Q with or without Sis1 overexpression as in Fig. 2. Representative images are shown, scale bar = 5 μm. Experiments were performed in triplicate, representative images shown. See also Supplementary Fig. 3a. **d** Increased solubility of Htt97Q in Sis1 overexpressing cells. The fraction of soluble material was analyzed in cell lysates fractionated into total (T), soluble (S) and pellet (P) fractions by centrifugation and analyzed by immunoblotting for mCherry (Htt97Q) and for PGK as loading control (see "Methods" section for details). Data were quantified by densitometry. Data represent mean ± SD from three independent experiments. **e** Htt97Q and Sis1 are immobile within inclusions. Htt97Q-mCherry (top) or endogenously tagged Sis1-GFP (bottom) were analyzed by fluorescence recovery after photo bleaching (FRAP) in cells expressing Htt97Q-mCherry with or without Sis1 overexpression. Representative images pre- and post-bleach are shown on the left, scale bar = 2.5 μm. Mean (solid lines) ± SE (traces) of fluorescence recovery of at least three experimental replicates are shown on the right. See Supplementary Fig. 3b for Htt20Q-mCherry control.

that Sis1 interacts with soluble polyQ oligomers, facilitating their coalescence into Sis1-containing condensates as a critical step in stress response induction. Ongoing polyQ synthesis is apparently required to generate the polyQ substrate recognized by Sis1.

**Sis1 enables efficient Hsp70 titration by polyQ.** Hsp40 proteins like Sis1 mediate substrate loading onto Hsp70 by stimulating the ATPase activity of Hsp70 proteins such as Ssa1 in the yeast

cytosol[76]. Mutation of the so-called HPD loop within the J-domain eliminates this activity[77,78] (Fig. 5a). The G/F-rich linker region of Sis1 is required for an as yet undefined, but essential cellular activity[78], while the C-terminal domain (CTD) is involved in substrate binding. Both Sis1 and DnaJB1, but not DnaJB6, contain a C-terminal dimerization domain (DD)[79] (Supplementary Fig. 2f). A Sis1 mutant lacking the DD preserved the activity to induce polyQ condensate formation and the stress response (Fig. 5b and Supplementary Fig. 5a). In contrast,

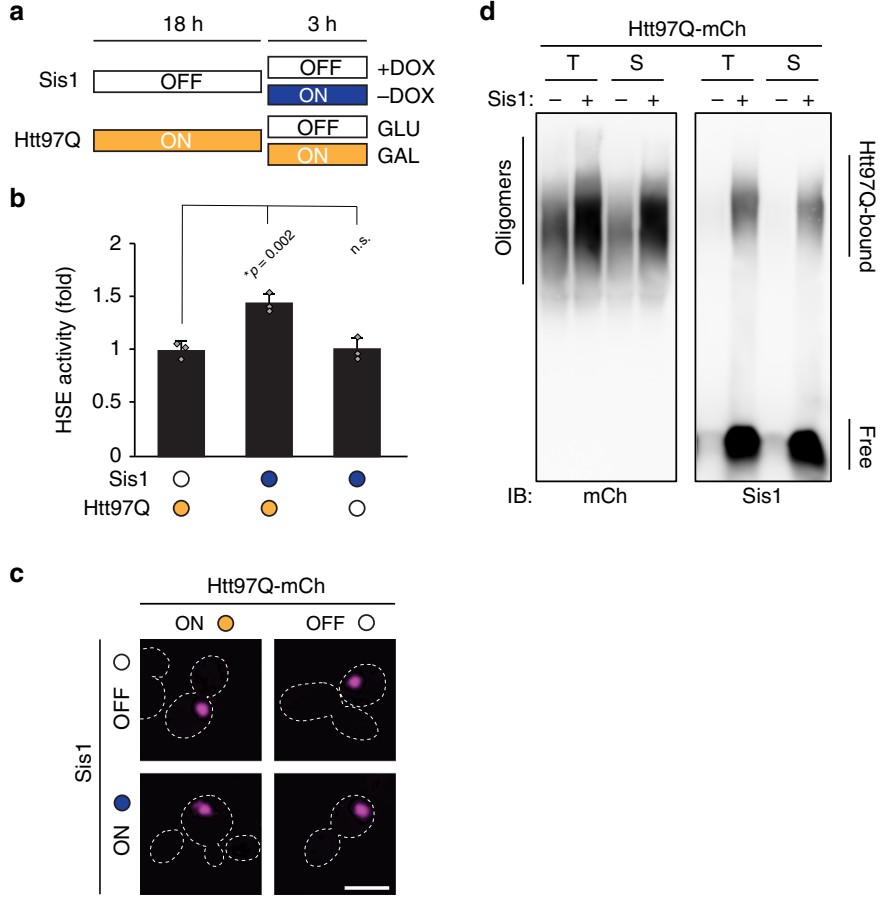

**Fig. 4 Sis1 acts on soluble polyQ oligomers. a** Experimental scheme for sequential expression of Htt97Q and Sis1. Htt97Q aggregates were allowed to form in the absence of Sis1 overexpression (OFF; white) for ~18 h. Samples were then split and subjected to continued (ON; orange) or repressed (OFF; white) Htt97Q expression (galactose or glucose mediated, respectively), and to normal (OFF; white) or overexpressed levels (ON; blue) of Sis1 (doxycycline mediated), for the final 3 h before harvesting and analysis. **b** Ongoing Htt97Q expression is required for Sis1 dependent HSR activation. $P_{HSE}$LacZ activities were measured in strains treated as in **a**. Htt97Q aggregates were allowed to form in cells by expressing Htt97Q for ~18 h. Htt97Q expression was then continued or repressed for 3 h without or with Sis1 overexpression (orange/empty and orange/blue dots, respectively). In a third reaction, Htt97Q expression was stopped by addition of glucose and Sis1 overexpressed for 3 h (empty/blue dots). Experiments were performed in triplicate. Data represent mean + SD from three independent experiments. *p values were determined by Dunnett's multiple comparisons t-test to Sis1 OFF control. n.s. not statistically significant. **c** Transient Sis1 overexpression does not remodel preexisting polyQ aggregates. Confocal microscopy was performed on cells during the conditions described in **a**. Scale bar = 5 μm. Experiments were performed in triplicate, representative images shown. **d** Overexpression of Sis1 increases the level of soluble polyQ oligomers. Total (T) or soluble (S) lysates from cells expressing Htt97Q with or without Sis1 overexpression (as in Fig. 3c) were treated with 2% SDS and analyzed by SDD-AGE (See "Methods" section for details). Immunoblotting was performed for mCherry (Htt97Q, left) or Sis1 (right). Representative blots of three independent experiments are shown. Also see Supplementary Fig. 4d.

deletion of the G/F region or the CTD caused the loss of both activities, and dense polyQ inclusions formed instead (Fig. 5b and Supplementary Fig. 5a), correlating with the failure of these Sis1 mutants to bind to polyQ oligomers on SDD-AGE (Supplementary Fig. 5b). Expression of the HPD mutant, Sis1-AAA[78], resulted in the formation of multiple polyQ foci rather than a single coherent condensate (Supplementary Fig. 5a). Sis1-AAA also failed to induce the stress response (Fig. 5b), but preserved the ability to form Sis1:polyQ oligomers (Supplementary Fig. 5b). Thus, both polyQ condensate formation and HSR induction require the functional interaction of Sis1 with Hsp70.

In addition to its role in protein folding, Hsp70 (primarily Ssa1 and Ssa2 in yeast) also functions as a regulator of the HSR by maintaining HSF1 in an inactive state[10,11,14,80]. Misfolded proteins are thought to titrate Hsp70 away from HSF1, thereby allowing HSF1 activation. Consistent with this model, overexpression of Ssa1 by ~50%, which is comparable to induction by heat stress[73] (Supplementary Fig. 5c), was sufficient to block the stress response to elevated temperature[81] (Supplementary Fig. 5d).

Increasing the level of Ssa1 also inhibited the polyQ mediated stress response in Sis1 overexpressing cells (Fig. 5c), demonstrating that Sis1 is acting through the same pathway. It is noteworthy that despite the inhibition of the stress response by Ssa1, polyQ condensates still formed upon Sis1 overexpression, confirming that this step occurs upstream of stress response activation (Fig. 5d). Note that overexpression of Ssa1 alone does not alter the appearance of the dense polyQ inclusions (Fig. 5d), although it can reduce the amount of SDS-insoluble polyQ protein[37].

To explore the interaction of Sis1 and Hsp70 with Htt97Q further, we performed in-cell chemical cross-linking with dithiobis(succinimidyl propionate) (DSP), followed by immunoprecipitation of Htt97Q from cell lysates. In control cells, low amounts of both chaperones were co-precipitated with Htt97Q but not with Htt20Q, consistent with previous reports[26,46] (Fig. 5e). The amount of co-precipitated Sis1 increased strongly upon Sis1 overexpression (Fig. 5e), consistent with the results from SDD-AGE analysis (Fig. 4d). Notably, the amount of Hsp70 (Ssa1/2) associated with Htt97Q also increased ~3.5-fold upon

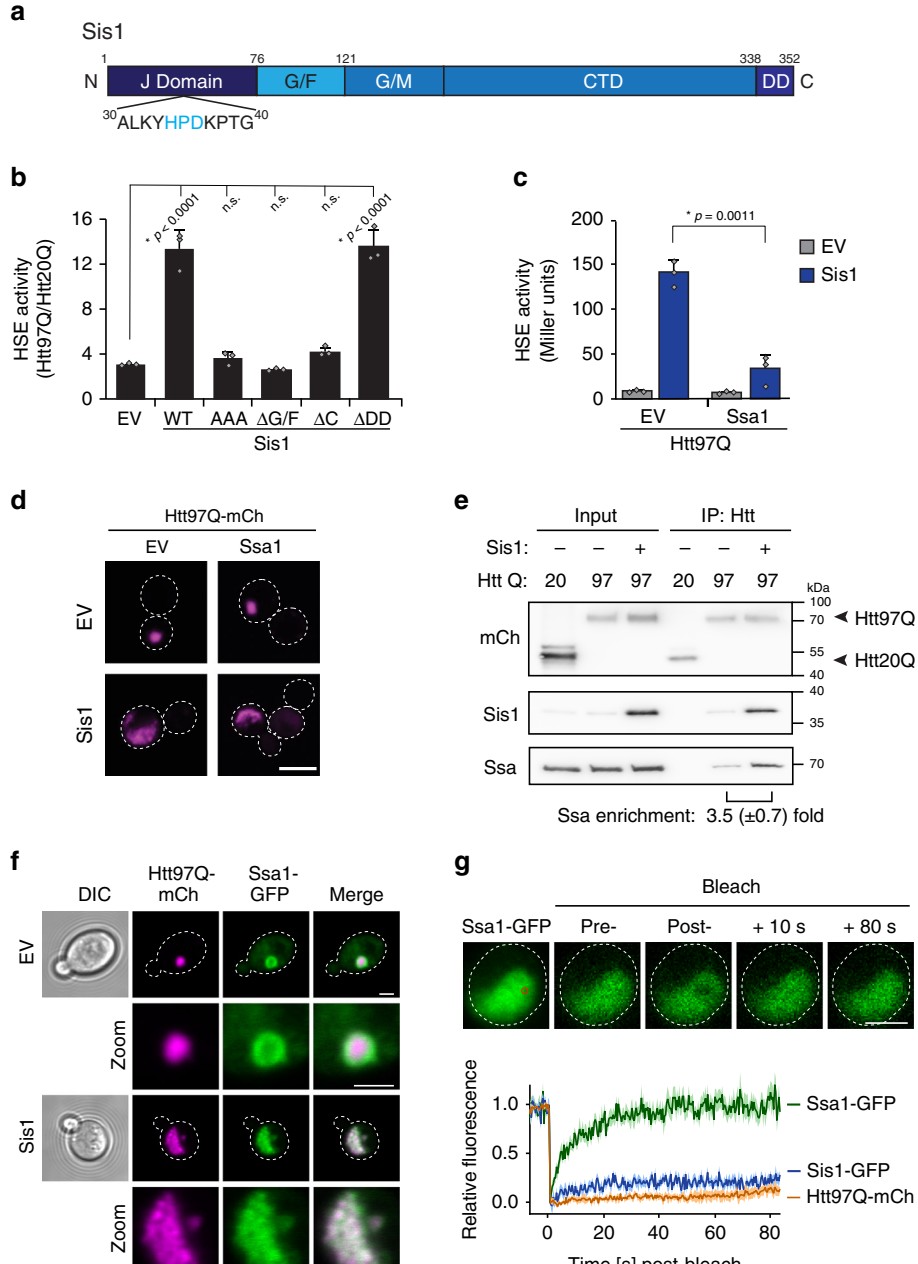

Sis1 overexpression (Fig. 5e). However, analysis by SDD-AGE of non-crosslinked samples showed no detectable association of Ssa1/2 with the Htt97Q oligomers (Supplementary Fig. 5e), in contrast to Sis1, suggesting that the interaction of Hsp70 with Htt97Q is more dynamic.

To visualize the interaction of Hsp70 with Htt97Q in cells, we used a yeast strain expressing a GFP-tagged copy of Ssa1. Ssa1-GFP co-localized with the Htt97Q aggregates, both with and without Sis1 overexpression (Fig. 5f). Notably, Sis1 overexpression markedly changed the topology of Ssa1 within the aggregates: In control cells, Ssa1-GFP was strongly reduced in the core of the inclusions, and was confined to the periphery. However, upon Sis1 overexpression, Ssa1 was distributed throughout the cloud-like Htt97Q condensate (Fig. 5f). This is in contrast to Sis1-GFP, which permeates both the polyQ inclusions formed at endogenous Sis1 levels and the condensates upon Sis1 overexpression (Fig. 3c). Strikingly, FRAP experiments indicated Ssa1-GFP in the condensates was mobile ($t_{1/2}$ of

recovery ~6 s), consistent with being in a liquid-like state, although Htt97Q and Sis1 remained relatively immobile (Figs. 3e and 5g).

In summary, the Sis1-mediated activation of the stress response can be resolved into the following steps: excess Sis1 interacts with soluble oligomers formed by newly-synthesized polyQ expanded protein. These Sis1-associated oligomers coalesce into cloud-like condensates in an Hsp70-dependent process. Hsp70 efficiently accumulates in these condensates, forming a mobile phase within an immobile Sis1-polyQ meshwork. Efficient recruitment of Hsp70 to polyQ leads to activation of the HSF1-mediated stress response.

**Sis1/DnaJB6 functions as a key regulator of the heat stress response.** Sis1 is of relatively low abundance compared to the other major cytosolic Hsp40, Ydj1 (~20,000 and ~110,000 molecules per cell, respectively), and compared to Hsp70 (~270,000 molecules for Ssa1 alone)[82]. Why would cells limit the

**Fig. 5 Sis1 recruits Hsp70 (Ssa1/2) to polyQ expanded Htt. a** Domain structure of Sis1. Sis1 is comprised of an N terminal J domain, containing an HPD motif critical for interaction with Hsp70, low-complexity G/F and G/M regions, a C-terminal substrate binding domain (CTD), and a C-terminal dimerization domain (DD). Numbers refer to amino acid residues. **b** Mutational analysis of Sis1 function in mediating HSR induction by Htt97Q. P$_{HSE}$LacZ activity was measured in cells co-expressing HA-tagged WT Sis1 or Sis1 mutants with Htt20Q or Htt97Q. EV, empty vector control; WT, Sis1 full length; AAA, HPD motif mutated to AAA; ΔG/F, Sis1 deleted for amino acids 77–121; ΔC, Sis1 amino acids 1–121; ΔDD, Sis1 amino acids 1–338. P$_{HSE}$LacZ activities upon Htt97Q expression are displayed as fold-change relative to Htt20Q expression. Data represent mean + SD from three independent experiments. *p values were determined by Dunnett's multiple comparisons t-test to EV control. n.s. not statistically significant. **c** Overexpression of Ssa1 blocks Sis1 mediated HSR induction in Htt97Q expressing cells. P$_{HSE}$LacZ activity was measured in cells grown at 30 °C expressing Htt97Q with or without Sis1 and Ssa1 co-overexpression, as indicated. Experiments were performed in triplicate. Data represent mean + SD from three independent experiments. *p values were calculated by unpaired, two-sided t-test. Also see Supplementary Fig. 5c. **d** Inhibition of HSR induction by Ssa1 overexpression does not block changes in polyQ aggregate morphology upon Sis1 overexpression. Confocal images of cells expressing Htt97Q as in **c** and Ssa1 with or without excess Sis1. Scale bar = 5 μm. Experiments were performed in triplicate, representative images shown. **e** Sis1 enables Ssa binding to Htt97Q. Cells expressing Htt20Q or Htt97Q with or without Sis1 overexpression were subjected to chemical crosslinking with dithiobis(succinimidyl propionate) (DSP) (see "Methods" section for details). Cell lysates were prepared and the Htt proteins immunoprecipitated with anti-Myc antibody. Input and eluate fractions were analyzed by immunoblotting for Ssa1/2, mCherry (Htt) and Sis1. Note that due to a high level of homology, the Ssa antibody used recognizes both Ssa1 and Ssa2. Representative results of three independent experiments are shown. The fraction of Ssa1/2 bound to Htt97Q was quantified by densitometry. Values represent mean ± SD. **f** Co-localization of Ssa1 with Htt97Q aggregates. Confocal microscopy was performed on cells containing a copy of *SSA1-GFP* expressing Htt97Q with or without Sis1 co-overexpression. Lower panels (Zoom) show magnified images. Scale bars = 2 μm. Experiments were performed in triplicate, representative images shown. **g** Ssa1 is dynamic within cloud-like Htt97Q condensates. Htt97Q condensates in cells expressing Ssa1-GFP with overexpression of Sis1 were analyzed by FRAP. Representative images prebleach and postbleach of Ssa1-GFP are shown (top). Mean (solid lines) ± SE (traces) of at least three experimental replicates of fluorescence recovery are graphed (bottom). FRAP of Htt97Q-mCherry and Sis1-GFP in condensates is shown as a reference (see also Fig. 3e). Scale bar = 2.5 μm.

level of Sis1, if more Sis1 enables the recognition of potentially dangerous protein aggregates? We considered the possibility that cells adjust Sis1 levels to allow this Hsp40 a more general role in the regulation of the HSR. Consistent with such a function, we found that elevating Sis1 strongly increased the response to heat-induced stress (Fig. 6a). The magnitude of the effect scaled with the level of Sis1 expression, as shown by comparing effects with Sis1 under the weak *CYC* promoter (~3-fold increase) and the strong *GPD* promoter, used in the previous experiments (~5-fold increase) (Fig. 6a and Supplementary Fig. 6a). Remarkably, potentiation of the HSR was specific to Sis1, as overexpression of Ydj1 did not lead to a pronounced difference in stress response induction (Supplementary Fig. 6b). Again, expression of the mammalian homolog DnaJB6 also enhanced the stress response (Supplementary Fig. 6b). The effect of Sis1 was observable at different temperatures (Supplementary Fig. 6c) and was not due to reporter infidelity or β-galactosidase misfolding (Supplementary Fig. 6d). Potentiation of the HSR required both the G/F and CTD domains of Sis1, as well as a functional interaction of the J-domain with Hsp70 (Supplementary Fig. 6e).

We noted however, that unlike for Htt97Q (Fig. 5c), over-expression of Ssa1 was no longer able to block the heat-induced stress response in cells with elevated Sis1 levels (Fig. 6b), indicative of dysfunction in stress response attenuation. More-over, while deletion of *SSA1* alone did not result in HSR induction at a normal growth temperature of 30 °C, overexpression of Sis1 activated the stress response in Δssa1 cells at 30 °C (Supplementary Fig. 6f). Thus, elevated Sis1 levels render yeast cells hypersensitive to stress. These results suggested that Sis1 functions as a limiting regulator for the HSR, and that Sis1 steady-state levels need to be low to prevent an overshooting of the stress response.

We next asked whether the function of Sis1 in regulating the HSR is conserved in mammalian cells. In contrast to Sis1, which is essential in yeast, deletion of the Sis1 homolog DnaJB6 is tolerated by mammalian cells in culture[62], allowing us to investigate the consequences of a loss of DnaJB6 function on stress regulation. Wild-type HEK293T cells or HEK293T cells deleted for DnaJB6 using CRISPR/Cas9[62] were maintained at 37 °C or subjected to a 1 h heat treatment at 43 °C. Deletion of DnaJB6 did not lead to major differences in the Hsp70 HspA1A

(which is expressed in HEK cells under non-stressed conditions) or HSF1 levels at the normal growth temperature of 37 °C (Supplementary Fig. 6g, h), suggesting that there were no gross changes to the proteostasis network. Strikingly, cells lacking DnaJB6 were substantially less efficient in upregulating the stress-dependent Hsp70B (HspA6)[83] or the small heat shock protein HspB1 (Hsp27) in response to heat stress (Supplementary Fig. 6g). Phosphorylation of HSF1, associated with stress response activity[11,84], occurred as in WT cells (Supplementary Fig. 6h), consistent with defective HSR induction being due to insufficient Hsp70 titration rather than a defect in HSF1 modification.

To recapitulate our studies in yeast, we next examined the effects of excess Hsp40s on the stress response in mammalian cells. Overexpression of DnaJB6 per se did not lead to an increase of HspA6 at 37 °C (Fig. 6c). However, when subjected to heat stress, cells overexpressing DnaJB6 showed strongly elevated levels of HspA6 compared to control cells (Fig. 6c). This enhancement of the HSR was relatively specific for DnaJB6, as overexpression of DnaJB1 showed a less pronounced effect. Furthermore, mutation of the J-domain of DnaJB6 (JB6-H31Q) resulted in a complete loss of HSR enhancement. Mutation of the unstructured S/T rich region (JB6-M3) (Supplementary Fig. 2f), previously implicated in the modulation of polyQ aggregation by DnaJB6[36,61], also showed less of an effect (Fig. 6c), consistent with the results for the unstructured G/F region in Sis1 (Fig. 5b, and Supplementary Figs. 5b, 6e). Taken together, we conclude that Sis1 and its mammalian homolog DnaJB6 are critical regulators of the cytosolic heat stress response, acting through a conserved mechanism.

**Sis1 recruits Ssa1 to heat-induced protein aggregates.** By employing the model protein Htt97Q, we were able to identify Sis1/DnaJB6 as potent regulators of the HSR in the absence of confounding environmental or toxicity effects. We next investigated whether Sis1/DnaJB6 potentiated the HSR to elevated temperature by a similar mechanism. Both Sis1 and Ssa1 have been reported to accumulate in aggregate foci during heat stress[73], suggesting that, in addition to Htt aggregates, these chaperones also recognize the aggregates of endogenous, heat-denatured proteins. Using the *SIS1-GFP* and *SSA1-GFP* strains described above (Figs. 3 and 5), we also observed the formation of

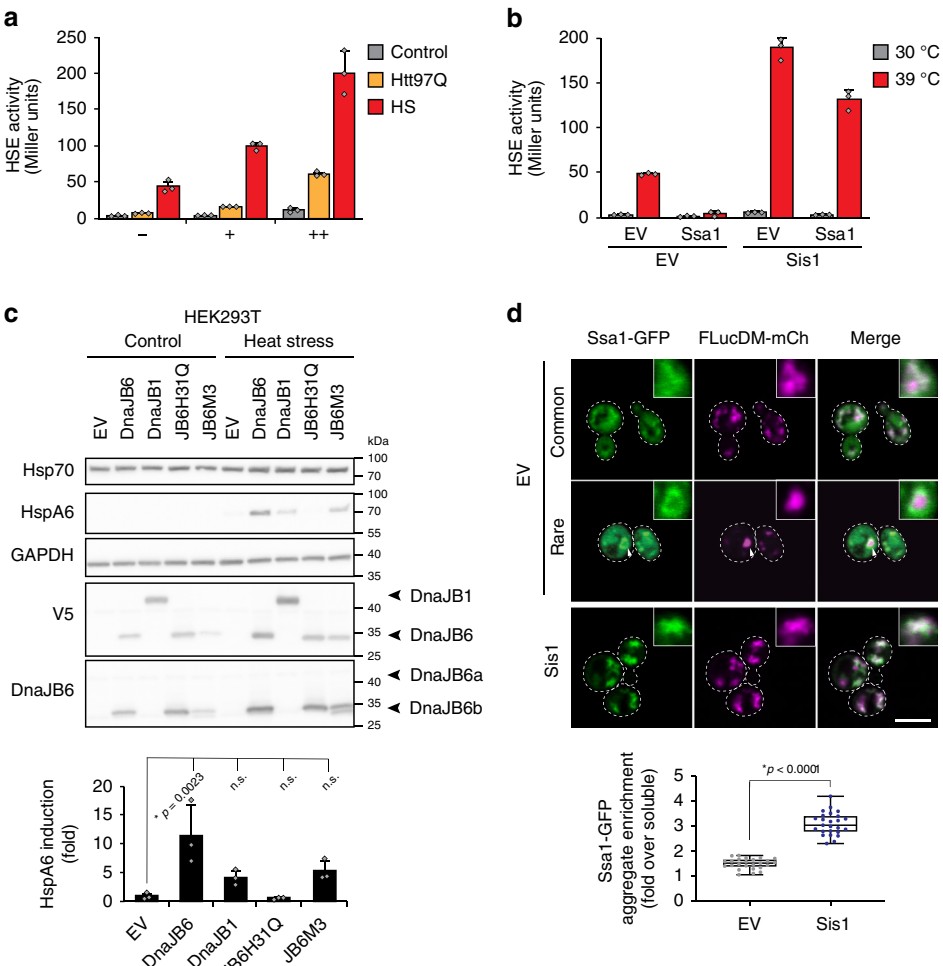

**Fig. 6 Sis1/DnaJB6 potentiates the cytosolic stress response. a** Sis1 enhances the HSR in yeast. $P_{HSE}$LacZ activity was measured in cells expressing Sis1 at endogenous (EV, −), mildly elevated ($P_{CYC}$Sis1, +), or highly elevated ($P_{GPD}$Sis1, ++) levels at 30 °C (gray bars) or after a mild heat stress of 1 h at 37 °C (red bars). Cells co-expressing Htt97Q at 30 °C were analyzed for comparison (orange bars). Data represent mean + SD from three independent experiments. **b** Ssa1 co-overexpression does not repress HSR enhancement in Sis1 overexpressing cells. $P_{HSE}$LacZ activity was measured in cells with or without Sis1 overexpression and with or without co-overexpression of Ssa1 at 30 °C (gray bars) or after a mild heat stress of 1 h at 39 °C (red bars). Data represent mean + SD from three independent experiments. **c** DnaJB6 potentiates the HSR in mammalian cell culture. Lysates from HEK293T cells transfected with a vector control, DnaJB6, DnaJB1, or DnaJB6 mutants H31Q, defective in interaction with Hsp70, or M3, which contains mutations within the unstructured S/T rich region were analyzed by SDS-PAGE and immunoblotting for the indicated proteins. Cells were grown at 37 °C and either maintained at 37 °C (Control) or exposed to heat stress for 1 h at 43 °C and recovery (Heat stress). Representative blots of three independent experiments are shown. Data represent mean + SD for three independent experiments. *$p$ values were calculated by Dunnett's multiple comparisons $t$-test to EV control. n.s. not statistically significant. **d** Co-localization of Ssa1 with FlucDM aggregates. Confocal microscopy was performed on cells containing a copy of *SSA1-GFP* under the *ADH* promoter and expressing FlucDM with or without Sis1 co-overexpression grown at 30 °C and incubated at 37 °C for 1 h. Representative images are shown. Arrow indicates example of rare ringed aggregate. Scale bars = 5 µm. Ratios of Ssa1-GFP in aggregates vs. soluble are quantified. Box plots represent median and 25th and 75th percentile, and whiskers minimal and maximal values. $n = 25$ cells. $p$ value was calculated by unpaired, two-sided $t$-test.

foci upon exposure to 37 °C (Supplementary Fig. 7a). To test whether excess Sis1 recruits Ssa1 to such aggregates, we expressed destabilized firefly luciferase (FlucDM), a model protein that misfolds and aggregates upon heat stress[85], tagged with mCherry under the *GAL1* promoter. FlucDM-mCh remained diffusely distributed at 30 °C, but formed aggregate foci at 37 °C that partially colocalized with Sis1, as detected in cells expressing endogenously tagged Sis1-GFP (Supplementary Fig. 7b). Expression of FlucDM-mCh resulted in a stronger HSR at the elevated temperature of 37 °C (Supplementary Fig. 7c), and this response was further enhanced upon overexpression of Sis1, despite similar levels of FlucDM expression (Supplementary Fig. 7d). We next expressed FlucDM-mCh in cells containing Ssa1-GFP with and without Sis1 overexpression. Ssa1-GFP

partially co-localized with the aggregates of FlucDM-mCh formed during heat stress at 37 °C (Fig. 6d). Overexpression of Sis1 enhanced the association of Ssa1-GFP with the FlucDM foci (Fig. 6d). Whereas at normal Sis1 levels, diffusely distributed Ssa1-GFP persisted after heat stress, the majority of Ssa1-GFP localized to aggregates upon Sis1 overexpression despite similar total levels of Ssa1-GFP (Fig. 6d and Supplementary Fig. 7e). Furthermore, in cells with normal Sis1, Ssa1-GFP occasionally formed "rings" around the FlucDM inclusions (Fig. 6d, arrow), similar to those observed for the dense Htt97Q aggregates (Fig. 5f). Such rings were not seen upon Sis1 overexpression. Together these results point to similar mechanisms of HSR potentiation by Sis1 in response to polyQ aggregates and heat induced protein aggregation.

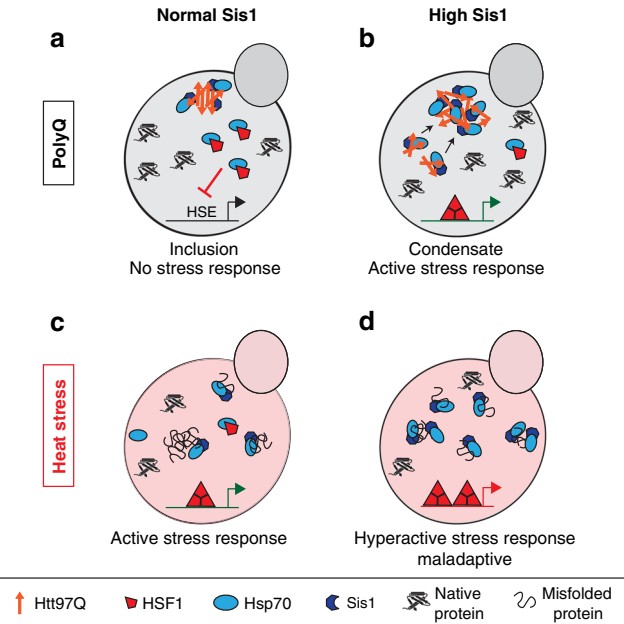

**Fig. 7 Hypothetical model for Sis1 function in regulating the stress response. a** Normal Sis1 levels. Htt97Q aggregates form dense inclusions that are inaccessible to Hsp70. HSF1 remains Hsp70-bound and inactive. HSE, heat shock element in the promoter. **b** Elevated Sis1 interacts with soluble polyQ oligomers and mediates their coalescence into cloud-like condensates that are permeable to Hsp70. As a result, Hsp70 is titrated away from HSF1, activating the HSR. Active HSF1 is shown as a trimer. **c** Normal Sis1 levels. During heat stress, Sis1 recruits Hsp70 to conformationally destabilized (misfolded) protein species. As a result, Hsp70 is titrated away from HSF1, activating the HSR. **d** Elevated Sis1 levels recruit an increased amount of Hsp70 to misfolded proteins during stress, resulting in a hyperactive, maladapted stress response.

## Discussion

Using a polyQ-based model of protein aggregation in yeast, we identified the essential Hsp40 chaperone Sis1 (and its mammalian homolog DnaJB6) as a critical regulator of the cytosolic stress response (HSR). Increasing the levels of Sis1 enables a robust induction of the HSR by polyQ-expanded huntingtin exon-1. Sis1 acts by directing the polyQ protein into a loosely packed condensate, avoiding formation of dense-core inclusions (Fig. 7a, b). Efficient recruitment of Hsp70 (Ssa1) into these permeable condensates results in HSR induction (Fig. 7b). In contrast, the inclusions formed at normal Sis1 levels are inaccessible to Hsp70, consistent with failure of stress response induction (Fig. 7a). Sis1 also sensitizes cells to heat stress, in a manner akin to a rheostat. This function is performed by DnaJB6 in mammalian cells (Fig. 7c, d). Importantly, elevating Sis1 leads to hyperactivation of the stress response (Fig. 7d), explaining why Sis1 levels are normally limiting.

Sis1 modulates the proteostasis system to allow recognition of aggregating polyQ protein by the stress response pathway. It thus functions as a positive regulator of the stress response that allows for aggregate-specific induction, in the absence of external environmental stressors. When present at increased levels, Sis1 transiently stabilizes soluble polyQ oligomers and recruits Hsp70 (Ssa1/2). Dependent on the functional cooperation of Sis1 with Hsp70, the polyQ oligomers then coalesce into an unusual cloud-like condensate with a ~4-fold lower polyQ density than the inclusions formed at normal Sis1 levels (Fig. 7a, b). Although the condensate readily dissociates upon cell lysis, it is not liquid-like, and thus distinct from the behavior of low complexity proteins in membrane-less compartments such as stress granules and P-granules[86–88].

Both Sis1 and Hsp70 (Ssa1) permeate the cloud-like polyQ condensates, but differ dramatically in dynamicity. This is consistent with previous reports that polyQ aggregates can sequester Sis1, preventing it from functioning in nuclear transport and quality control[26]. In contrast, the dynamic sequestration of Hsp70 is a functional interaction that leads to a productive stress response, thus highlighting the importance of dynamicity as a determinant of biological outcome[32]. Whereas polyQ and Sis1 are essentially immobile, Ssa1 shows highly mobile phase behavior. Thus, Ssa1 forms a liquid-like phase within the immobile Htt97Q meshwork, presumably making transient interactions with Sis1 and the polyQ protein during ATP-dependent cycling.

Efficient recruitment of Hsp70 into the condensate is critical for HSR induction by titrating Hsp70 away from HSF1, consistent with the prevailing model of HSR regulation[16] (Fig. 7b). This conclusion rests on the following lines of evidence: (i) In contrast to the condensates, the inclusions formed at normal Sis1 levels have a dense core that is largely impermeant to Ssa1, limiting Ssa1 recruitment. (ii) Overexpression of Ssa1 prevented the Sis1-mediated stress response to polyQ, apparently by maintaining HSF1 in its Hsp70-bound, inactive state, but did not interfere with condensate formation.

In addition to a functional requirement for Hsp70 interaction, the ability of Sis1 to modulate polyQ aggregation is dependent on both the G/F low-complexity region and the C-terminal substrate binding domain. Given the widespread involvement of low-complexity sequences in phase separation phenomena[88–91], it seems plausible that the G/F region has a direct role in polyQ condensate formation. While all type I and II Hsp40s have a structurally disordered G/F domain[59], in Sis1 this region is uniquely critical for the essential functions of Sis1[78,92]. Specific inter-domain interactions of the G/F region may distinguish Sis1 from other G/F containing Hsp40s[78,93]. Similarly, mutations in the G/F domain of DnaJB6 are associated with a type of limb-girdle muscular dystrophy[94], underscoring the critical role of this domain. Interestingly, the C-terminal domain of DnaJB6 contains an additional S/T-rich sequence and a histone deacetylase binding domain, which has been implicated in the superior aggregation prevention capacity of this chaperone for polyQ proteins[36,61].

If more Sis1 sensitizes the stress response pathway towards potentially toxic protein aggregates, why would cells maintain Sis1 at a limiting level, precluding such a response? Examining the heat-induced stress response under conditions of excess Sis1 (and limiting DnaJB6 in mammalian cells) provided an answer to this question: Sis1/DnaJB6 has a general role in regulating the HSR (Fig. 7c, d). Elevating Sis1 in yeast and DnaJB6 in mammalian cells sensitizes cells to heat stress, with the magnitude of the response scaling with Sis1 levels. Cells limit Sis1 in order to maintain a sensitive and dynamic heat stress-sensing pathway and avoid an overshooting, maladaptive response. The discovery of this additional regulatory element also provides an explanation as to how the stress response can be sensitive to even minute changes in the environment despite the abundance of downstream factors such as Hsp70.

Regulation of the HSR depends on the G/F region, the CTD and J-domain of Sis1, suggesting a similar underlying mechanism in titrating Hsp70 away from HSF1 as in the response to polyQ aggregation. In support of this possibility, transient protein aggregation in a chaperone-regulated manner has been observed as one of the first consequences of heat-induced protein mis-folding in yeast[95]. Recruitment of Hsp70 to these so-called Q-bodies may induce the stress response and this effect could be enhanced when Sis1 is elevated[96]. This is consistent with our observations that elevating Sis1 enhances the recruitment of Ssa1

to heat-induced aggregates of the model protein mutant luciferase. Amyloid aggregates typically do not accumulate in Q-bodies in the absence of heat stress[95], and therefore may escape detection. Thus, the polyQ condensates observed at elevated Sis1 levels may be considered the functional Q-body equivalent for this amyloidogenic protein.

Shuttling between the cytosol and the nucleus[26] likely endows Sis1 with additional regulatory potential. Recent studies suggest that nuclear Sis1 is critical in loading Hsp70 (Ssa) onto HSF1, maintaining the transcription factor in a repressed state under non-stress conditions[97,98]. Binding of Sis1 to protein aggregates, presumably including metastable nuclear proteins, would therefore not only recruit Hsp70 away from HSF1, but also delay Hsp70 re-binding to HSF1 and prolong the stress response. The existence of a nuclear isoform of DnaJB6[63] suggests a similar regulatory function for this Sis1 homolog in mammalian cells. It will be interesting in this context whether nuclear DnaJB6 has a role in mediating the rapid, stress-induced transfer of Hsp70 into the nucleolus[99], which might serve as an efficient mechanism to remove Hsp70 from HSF1.

In yeast, the endogenous amyloidogenic proteins (yeast prions) are not generally toxic[100]. However, polyQ aggregation is associated with toxicity in mammalian cells and overexpression of DnaJB6 has been shown to be protective[36,60,61]. On the other hand, overexpression of DnaJB6 can itself lead to toxicity in primary neurons[101], suggesting a fine balance is required for proteostatic health. In support of this idea, it has been shown that DnaJB6 levels decrease during differentiation of neuronal cells, leaving them more susceptible to the toxic effects of polyQ aggregates[62] and presumably less responsive to other forms of proteotoxic stress as well. Finding ways to counteract this decline may be protective against polyQ diseases and other degenerative pathologies associated with protein aggregation.

## Methods

**Molecular cloning**. Plasmids and primers used in this study are listed in Supplementary Tables 4 and 5, respectively. Standard DNA cloning was performed using restriction digest and DNA ligation using T4 DNA ligase (New England Biolabs). PCR was performed using Q5 DNA Polymerase (New England Biolabs). Gibson Assembly cloning was performed using the Gibson Assembly® Cloning Kit (New England Biolabs).

The nuclear reporter plasmids pRS306P$_{GPD}$-GST-GFP-NLS and pRS306P$_{GPD}$-GST-mCh-NLS were generated by three-way ligation between vector pRS306P$_{GPD}$[102] digested with XbaI and SalI, a DNA fragment encoding GST digested with XbaI and BamHI, and a DNA fragment encoding GFP or mCherry with a C-terminal nuclear localization signal (NLS) digested with BamHI and SalI (New England Biolabs). The GST fragment was generated by PCR with pGEX-2T-Tev as a template with primers 5 GST XbaI and 3 GST GS3 BHI. The GFP-NLS fragment was generated by PCR with pFA6A-GFP(S65T)-KanMX6 as a template and primers 5 GFP BHI and 3 GFP NLS SalI. The mCh-NLS fragment was generated by PCR with pYes2-myc-20QmCh as a template and primers 5 mCh BHI and 3 mCh NLS SalI.

A DNA fragment encoding HSF1 was amplified from pRS426Hsf1[103] using primers 5 HSF1 EcoRI and 3 HSF1 XhoI and inserted into vector pRS414P$_{GPD}$[102] to generate pRS414P$_{GPD}$HSF1.

A DNA fragment encoding the LacZ gene was amplified from plasmid Ssa3-LacZ-Leu[49] by PCR using primers 5 LacZ BamHI and 3 LacZ SalI and cloned into a similarly digested pRS425P$_{GAL}$ (ATCC) to generate pRS425P$_{GAL}$LacZ.

Plasmids pESCLeu-20QmCh and pESCLeu-97QmCh were generated by ligating the BamHI and MluI (New England Biolabs) restriction digested insert from pYES2-Myc-20QmCh or pYes2-Myc97QmCh[26] into a similarly digested pESCLeu vector (Agilent).

The heat-stress inducible promoter and LacZ region were amplified from plasmid Ssa3-LacZ-Leu[49] using primers 5 pHSE BglII and 3 LacZ SalI, digested with BglII and SalI (New England Biolabs) and cloned into a BamHI and SalI digested pRS313 to generate pRS313P$_{HSE}$LacZ.

The plasmids used in the overexpression screen were isolated from the Yeast ORF Collection (Dharmacon) and contain the chaperones within a multicopy, 2-micron, galactose-inducible yeast expression vector (BG1805).

YDJ1 was amplified from the genome of yeast strain YPH499[104] using the primers 5 Ydj1 SpeI and 3 Ydj1 BHI. The resulting fragment was subcloned into the pRS414P$_{GPD}$ vector of a SpeI and BamHI digested pRS414P$_{GPD}$Sis1[105] to generate pRS414P$_{GPD}$Ydj1.

DNA fragments encoding V5 tagged DnaJB1 or DnaJB6b were amplified from pcDNA3 V5 DnaJB1 or pcDNA3 V5 DnaJB6[28,36] using primers 5 V5 SpeI and 3 DnajB1 XhoI or 5 V5 SpeI and 3 DnajB6b XhoI, respectively, and cloned into a SpeI and XhoI (New England Biolabs) digested pRS414P$_{GPD}$ vector[102].

The mid-length polyQ plasmid pYES2-Myc-48QmCh was created by first isolating the 48Q Htt fragment from pBacMam2-DiEx-LIC-C-flag_huntingtin_full-lengthQ48[106] via restriction digest, first with PvuI and XhoI, then with StuI and TfiI (New England Biolabs). The restriction product was integrated via Gibson Assembly into the vector created by PCR using pYES2-Myc-97QmCh as a template with primers PolyQBB_fw and int.PolyQrev.

SIS1 was amplified from pRS414P$_{GPD}$Sis1[105] using primers 5 Sis1 BamHI and 3 Sis1 Not1 and inserted into the multiple cloning site of pCM184[107].

A DNA fragment encoding HA-tagged Sis1 and Ydj1 were amplified from plasmids pRS414P$_{GPD}$Sis1 and pRS414P$_{GPD}$Ydj1 using primers 5 Spe1 HASis1 and 3 Sis1 BHI or 5 SpeI HAYdj1 and 3 Ydj1 BHI, respectively, and subcloned into a BamHI and SpeI digested pRS414P$_{GPD}$Sis1 vector.

A DNA fragment encoding HA-tagged WT Sis1, Sis1 with the HPD motif of the J-domain mutated to AAA, or Sis1 lacking the G/F region from amino acids 71-121, were subcloned with their promoters from pRS415P$_{GAL}$-HA-Sis1, pRS415P$_{GAL}$-HA-Sis1(AAA), or pRS415P$_{GAL}$-HA-Sis1ΔG/F[26] into a pRS414 vector[104] to create pRS414P$_{GAL}$-HA-Sis1, pRS414P$_{GAL}$-HA-Sis1AAA, and pRS414P$_{GAL}$-HA-Sis1ΔG/F, respectively. A DNA fragment encoding HA plus the first 121 or 338 amino acids of Sis1 was created by PCR of pRS414P$_{GAL}$-HA-Sis1 with primers 5 SpeI HA-Sis1 and 3 Sis1-121 BHI or 5 SpeI HA-Sis1 and 3 Sis1-338 BHI and cloned to replace the WT Sis1 in pRS414P$_{GAL}$-HA-Sis1 to create pRS414P$_{GAL}$-HA-Sis1ΔC or pRS414P$_{GAL}$-HA-Sis1ΔDD, respectively.

SSA1 from pRS426Ssa1[37] was inserted into the BamHI and XhoI sites of PRS413P$_{GPD}$ to create plasmid PRS413P$_{GPD}$Ssa1.

To create plasmid pRS405P$_{ADH}$-Ssa1GFP, first an SSA-GFP fusion was created by amplifying SSA1 from pRS413P$_{GPD}$-Ssa1 with primers pRS_Ssa1_fw and Ssa1_GS_rev and GFP from pFA6A-GFP(S65T)-KanMX6 with primers GS_GFP_fw and GFP_PRS_rev. The SSA and GFP fragments were subsequently fused by PCR using primers pRS_Ssa1_fw and GFP_pRS_rev. This SSA-GFP fusion was then inserted via Gibson assembly into the backbone of plasmid pRS405P$_{ADH}$ which was amplified using primers pRS_MCS_fw and pRS_MCS_rev.

The GPD promoter from pRS414P$_{GPD}$Sis1[105] was replaced with a CYC promoter by subcloning with SacI and SpeI (New England Biolabs) digested pRS413P$_{CYC}$[102] to generate pRS414P$_{CYC}$Sis1.

pYES2-FlucDM-mCh was created by first subcloning an XhoI and XmaI (New England Biolabs) generated DNA fragment encoding FlucDM from pCIneo-FlucDM[85] to a similarly digested mCherry-N1 plasmid (Clontech). The FlucDM-mCherry fusion was subcloned into the pYES2 vector (Invitrogen) using KpnI and XbaI restriction sites (New England Biolabs).

**Yeast strains**. Yeast strains used in this study are listed in Supplementary Table 6. The GFP-NLS strain was created by transforming YPH499[104] with BsmI (New England Biolabs) cut plasmid pRS306P$_{GPD}$-GST-GFP-NLS and selection on media lacking uracil.

The [pin⁻] strain was created by repeated passaging of WT YPH499 cells on YPD containing 3 mM GDN (guanidine HCl). Prion status was confirmed by transient expression of Rnq1-GFP and examination for isolates with soluble Rnq1-GFP by microscopy.

The SSA1-GFP strain was created by transforming strain YPH499 with BstEII (New England Biolabs) cut plasmid pRS405P$_{ADH}$Ssa1GFP and selection on media lacking leucine.

The SIS1-GFP strain was created by transforming a PCR generated cassette using pFA6a-GFP(S65T)-KanMX6[108] as a template with primers 5 Sis1-GFP and 3 Sis1-GFP into strain YPH499. Transformants were selected by growth on media containing 300 µg/ml G418 and confirmed by genomic PCR.

The mCherry-NLS strains were created by transforming the SSA1-GFP or SIS1-GFP strains with BsmI (New England Biolabs) cut plasmid pRS306P$_{GPD}$-GST-mCh-NLS and selection on media lacking uracil.

**Growth and heat treatment**. Yeast strains were cultured at 30 °C, unless stated otherwise. Heat treatments were conducted in a pre-warmed 37 °C or 39 °C incubator, whereas treatment at 50 °C was performed in a 50 °C shaking heat block. Cultures were grown in synthetic complete media lacking amino acids necessary for plasmid selection and supplemented with either 2% glucose or 2% raffinose and 2% galactose. Experimental comparisons were made between identical growth conditions unless indicated otherwise. Cultures were maintained for >18 h with dilution and harvested during log phase.

HEK293T and DnaJB6 knockout cells[62] (Supplementary Table 6) were cultured in DMEM supplemented with 10% Fetal Bovine Serum, Penicillin and Streptomycin in a 37 °C humidified incubator at 5% $CO_2$. Passaging was performed twice weekly. Transient transfections were performed using 1 µg of DNA and PEI transfection reagent (1:6 ratio). Heat shock experiments were performed for 1 h at 43 °C in a water bath. Samples were collected after ~24 h recovery at 37 °C, unless otherwise indicated.

**Preparation of cell extracts**. Yeast pellets were resuspended in 500 μl lysis buffer (25 mM Tris pH 7.5, 50 mM KCl, 10 mM MgCl$_2$, 1 mM EDTA, 5% glycerol, 0.5% Triton X-100) supplemented with protease inhibitors (cOmplete mini, EDTA-free Protease Inhibitor Cocktail, Roche). Cells were lysed by bead-milling with acid washed glass beads in a MP Beadbeater24 (2 × 20 s at 6.0 m/s with intermittent cooling on ice). Extracts were cleared from cell debris by centrifugation at 500 × g for 5 min at 4 °C. Protein levels were normalized using the Bio-Rad Protein Assay (Bio-Rad). Samples containing polyQ expanded Htt were first precipitated with 25% trichloroacetic acid (TCA) before dissolution with formic acid at 37 °C and preparation for SDS-PAGE[109].

Mammalian cell pellets were resuspended in RIPA buffer (25 mM Tris pH 7.4, 150 mM NaCl, 1 mM MgCl$_2$, 1% NP40, 1% sodium deoxycholate, 0.1% SDS) supplemented with protease inhibitors (Roche) and DENARASE (c-LEcta, 50 U/ml) and vortexed on ice for 1 h. Protein levels were normalized using the DC Protein Assay (Bio-Rad).

**Cell fractionation**. Total yeast cell lysates were separated into soluble and pellet fractions by centrifugation at 15,000 × g for 15 min at 4 °C. After collection of the soluble supernatant fraction, pellets were washed with lysis buffer (2 × 15,000 × g for 5 min at 4 °C).

**SDS-PAGE**. Protein samples were suspended in sample buffer (50 mM Tris pH 6.8, 8% glycerol, 2% SDS, 20% β-mercaptoethanol, containing bromophenol blue), heated to 95 °C for 3 min and separated by electrophoresis on 10 or 12% self-made Tris-glycine poly-acrylamide (Serva) or TGX FastCast acrylamide solution (Bio-Rad) gels using Laemmli (25 mM Tris, 192 mM glycine, 0.1% SDS) running buffer at 30 mA/gel.

**Filter retardation assay**. Filter retardation assays for the detection of total and SDS resistant polyQ aggregates were performed as previously described[26,37,45]. Cell lysates were incubated with 125 U benzonase (Novagen) or smDNase (Max Planck Institute for Biochemistry core facility) with rotation for 1 h at 4 °C. Samples were then incubated with or without SDS buffer (2% SDS, 50 mM DTT) before being loaded onto a pre-wetted (0.1% SDS) 0.2 μm pore size cellulose acetate membrane in a Hoefer slot-blot apparatus. Membranes were subsequently washed four times with 0.1% SDS before immunoblotting.

**SDD-AGE**. Semi-denaturing detergent agarose gel electrophoresis was performed as previously described[75] with a modified lysis buffer (25 mM Tris pH 7.5, 50 mM KCl, 10 mM MgCl$_2$, 1 mM EDTA, 5% glycerol, 0.5% Triton X-100). Yeast extracts were incubated at 30 °C for 7 min in SDD-AGE sample buffer (50 mM Tris pH 6.8, 8% glycerol, 2% SDS, supplemented with bromophenol blue). Samples were then run on an agarose gel (1.5% agarose, 0.1% SDS, in 25 mM Tris, 192 mM glycine) in running buffer (25 mM Tris, 192 mM glycine, 0.1% SDS) at 160 V for 1 h at 4 °C.

**Immunoblotting**. Proteins were transferred from poly-acrylamide gels to nitrocellulose membranes in transfer buffer (25 mM Tris, 192 mM glycine, 20% methanol) at 110 V for 1 h using the Mini Trans-Blot Cell (Bio-Rad) or for 10 min using the Trans-Blot Turbo Transfer System (Bio-Rad). Proteins were transferred from agarose gels (SDD-AGE) to nitrocellulose membranes in transfer buffer containing 0.01% SDS at 7 V for 14 h using the Genie Blotter System (Research Products International). Membranes were washed with TBST (10 mM Tris-HCl pH7.5, 150 mM NaCl, 0.1% Tween-20), blocked with 4% powdered milk in TBST, and incubated with primary antibodies overnight at 4 °C. Information regarding the primary antibodies used in this study can be found in Supplementary Table 7. After washing with TBST, membranes were incubated with horseradish peroxidase (HRP)-conjugated secondary antibodies for 1 h and washed again. Imaging was performed using Luminata Classico substrate and the ImageQuant LAS 4000 mini detector or Bio-Rad ChemiDoc imaging system. Densitometric analyses were performed using LICOR Image Studio Lite or Image Lab (Bio-Rad) software.

**Immunoprecipitation after crosslinking**. Yeast cells were pelleted and resuspended in crosslinking (XL) buffer (1.2 M sorbitol, 5 mM EDTA, 0.1 M KH$_2$PO$_4$/K$_2$HPO$_4$ pH 7.5). DSP solution (50 mM dithiobis(succinimidyl propionate) in DMSO) was added to 1 mM final concentration and samples were incubated at 30 °C for 20 min before quenching by addition of 0.1 volume 1 M Tris pH 7.5 for 15 min. Lysates were prepared as above and soluble fractions isolated by centrifugation at 15,000 × g for 15 min at 4 °C. Thousand microgram of protein (determined by Bradford assay) were diluted in lysis buffer, total fractions removed, and remaining samples incubated with 50 μl Milltenyi μMACS magnetic anti-Myc beads for 1 h at 4 °C. Samples were passed over the magnetic columns and washed (4× lysis buffer with 1% Triton X-100 and 2 × 20 mM Tris-HCl pH 7.5) before elution with hot SDS sample buffer.

**Confocal imaging**. Mid-to-late log phase yeast cells (0.5–1 OD$_{600}$) were adhered to the chamber of a concanavalin A coated μ-Slide (Ibidi, 80626) for 5 min at room temperature. Cells were washed three times with culture media. Imaging was done with an Olympus (Tokyo, Japan) FV1000 confocal microscope setup equipped with

an Olympus PLAPON 60×/NA1.42 oil immersion objective. The GFP fluorophore was detected using an excitation wavelength of 488 nm and emission of 505–540 nm. The mCherry fluorophore was detected using an excitation wavelength of 559 nm and emission of 575–675 nm. Image analysis was carried out in Fiji[110].

**Analysis of aggregate density**. Cells were prepared as described above and imaged with an Olympus (Tokyo, Japan) FV1000 confocal microscope setup equipped with an Olympus PLAPON 60×/NA1.42 oil immersion objective using the same nonsaturating acquisition settings for all samples. Image analysis was carried out in Fiji[110]. To analyze the fluorescent density of aggregates, circular regions within the aggregate were selected and the average fluorescence signal was compared between samples. To obtain the total cellular fluorescence signal, the integrated fluorescence density over the whole area of the analyzed cells was calculated.

**Analysis of Ssa1 enrichment**. Cells were prepared as described above and imaged with an Olympus (Tokyo, Japan) FV1000 confocal microscope setup equipped with an Olympus PLAPON 60×/NA1.42 oil immersion objective using the same non-saturating acquisition settings for all samples. Image analysis was carried out in Fiji[110]. Regions of interest were selected outlining the FLuc aggregate as well as a large area in the cell not containing any aggregate (soluble). The enrichment of Ssa1 in aggregates over soluble signal was calculated as the ratio between the average fluorescence of Ssa1 in these two regions. To obtain the total cellular fluorescence signal, the integrated fluorescence density over the whole area of the analyzed cells was calculated.

**FRAP analysis**. Samples were prepared as described above. In vivo fluorescence recovery after photobleaching (FRAP) experiments were carried out at MPIB Imaging Facility (Martinsried, Germany) on a ZEISS (Jena, Germany) LSM780 confocal laser scanning microscope equipped with a ZEISS Plan-APO 63×/NA1.46 oil immersion objective. Circular regions of constant size were bleached after 20 frames and monitored for at least 80 s (0.38 s/frame) for fluorescence recovery in a single focal plane. Image analysis was carried out in Fiji[110]. Fluorescence intensity data were corrected for photobleaching and normalized to the average fluorescence intensity before (1) and after (0) bleaching (relative fluorescence), except the experiment including soluble material, which was only normalized pre-bleach due to the rapid recovery (reported as fluorescence intensity).

**β-Galactosidase activity measurements**. LacZ expression was measured using a standard ß-galactosidase assay[103]. Briefly, cell pellets were resuspended in 700 μl of Z buffer (60 mM Na$_2$HPO$_4$, 40 mM NaH$_2$PO$_4$, 10 mM KCl, 1 mM MgSO$_4$, 50 mM ß-mercaptoethanol) supplemented with 50 μl of 0.1% SDS and 50 μl of chloroform, vortexed and incubated for 5 min at 30 °C. Two hundred microliter of substrate (4 mg/ml ONPG (2-Nitrophenyl β-D-galactopyranoside, Sigma-Aldrich) in Z buffer) was added with further incubation (~3–13 min) before quenching with 350 μl 1 M Na$_2$CO$_3$, clearance by centrifugation at 500 × g, and spectrophotometric measurement at A$_{420}$. Activity was calculated in standard Miller Units. 1 Miller Unit = 1000 × (A$_{420}$ / (O.D.$_{600}$ × ml × t$_{min}$)).

**mRNA sequencing**. Cells were harvested in triplicate and sent to Novogene Co., Ltd. (Hong Kong) for RNA extraction, library preparation, Illumina paired-end sequencing, mapping, and bioinformatics analysis. Briefly, 1 μg RNA per sample was used for library generation using NEB Next Ultra RNA Library Prep Kit for Illumina (NEB). Library preparations were subjected to Illumina paired-end sequencing (150 bp, 10 million reads). Reads were indexed using Bowtie v2.2.3 and aligned to the reference genome using TopHat v2.0.12. Read numbers were counted using HTSeq v0.6.1 and gene expression was calculated using FPKM (Fragments per Kilobase of transcript sequence per Million base pairs sequenced) method. The differentially expressed genes between samples were calculated using the DESeq R package and the resulting p values were adjusted using the Benjamini and Hochberg false discovery rate approach. Differentially expressed genes are defined as having an adjusted p value less than 0.05. Gene ontology (GO) enrichment analysis of the differentially expressed genes was performed using the GOseq R package. Significance was corrected for gene length bias and adjusted p values less than 0.05 indicate a significantly enriched term.

**Statistical analysis**. Error bars represent standard deviation from at least three experiments. Significance was determined for two-sample comparisons using the unpaired t-test function with a threshold of two-tailed p values less than 0.05. Significance among multiple conditions were determined using ANOVA and Dunnett's multiple comparisons test with a threshold of p < 0.05. Calculations were performed using Graphpad software.

**Reporting summary**. Further information on research design is available in the Nature Research Reporting Summary linked to this article.

## Data availability

The transcriptome data discussed in this publication have been deposited in NCBI's gene expression omnibus and are accessible through GEO series accession number GSE151215. The string database is available at https://string-db.org (v 11, organism S. cerevisiae). All other data supporting the findings of this study are available within the manuscript. Source data are provided with this paper.

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

## Acknowledgements

We thank Johannes Buchner (Technical University of Munich), Elizabeth Craig (University of Wisconsin—Madison) and Ineke Braakman (University of Utrecht) for

providing antibodies, Harm Kampinga (University Medical Center Groningen) for providing plasmids and cell lines, Eduardo de Mattos and Arun Thiruvalluvan (University Medical Center Groningen) for assistance with the DnaJB6 cell line, and Yury Kukushkin and Sae-Hun Park for preliminary data. We thank Lucas Cairo, Gopal Jayaraj, and Sae-Hun Park for critically reading the manuscript. FRAP experiments were performed at the Max Planck Institute of Biochemistry Imaging Core Facility. This research has received funding from the European Commission (FP7 GA ERC-2012-SyG_318987–ToPAG) and the Munich Cluster for Systems Neurology (SyNergy). C.L.K. acknowledges funding by the Alexander von Humboldt Foundation (Postdoctoral Fellowship 3.1-USA/1162753 HFST-P). M.H.M.G. was supported by a DFG fellowship through the Graduate School of Quantitative Biosciences Munich (QBM).

## Author contributions

C.L.K. planned and performed most experiments and analyzed the data. M.H.M.G. assisted with experiments and performed the imaging and FRAP analysis. F.U.H. conceived the project and participated in data interpretation together with the other authors. C.L.K and F.U.H. wrote the paper with contributions from M.S.H.

## Funding

## Competing interests

The authors declare no competing interests.
