## [Peer Review File · Nature Communications]

REVIEWER COMMENTS

Reviewer #1 (Remarks to the Author):

In this manuscript Klaips et al. establish that the J-domain protein Sis1 is a limiting component for Hsf1 to induce stress-gene expression in response to aggregating polyQ. Overproducing Sis1 converts the dense, Hsp70 inaccessible PolyQ inclusions to a permeable meshwork that recruits Hsp70, thus triggering the heat shock response via liberation of Hsf1. In addition, the main findings are recapitulated using the human Sis1 homolog DnaJB6, recombinantly expressed in yeast as well as investigated in the mammalian background.

This study provides conceptual insight into how the proteostasis system is organized to detect specific misfolded species. The study suggests that Sis1 functions as a rheostat to set the sensitivity threshold of the heat shock response. Overall, the manuscript is well written and the conclusions have solid experimental support.

Major concerns

1. A major claim in the manuscript is that Sis1 functions as a rheostat of the Hsf1-driven heat shock response. The study provides convincing data in support of the notion that physiological Sis1 levels are limiting for Hsf1 activation via Htt97Q-mediated titration of Hsp70. Yet, the rheostat concept implies that physiological changes of Sis1 levels modulate Hsf1 activity. This key aspect is not developed in the manuscript, the experiments instead solely rely on genetic manipulation of Sis1 levels. This is a major limitation of the study and raises the question whether Sis1 levels really do change under different stress/growth conditions and if this translates into transcriptional rheostat. Such a demonstration would strengthen the support for the claim of a Sis1 rheostat function.

2. The transcriptome analysis (lines 177-203, Fig 2c-d and Suppl tables) is not clearly outlined and needs to be more explicit. Is it simply Hsf1 target genes that are induced by Sis1 overexpression in the Htt97Q condition? What other regulons are activated by Htt97Q expression (Msn2/4?)? Probably volcano plots provide the most transparent method of presenting the differentially expressed genes (DEs). The DEs can be color coded in the volcano plots based on what regulons they belong to. By eyeballing, the Hsf1 regulon appears to be dominating the DEs in the comparison Htt97Q with or without overexpression Sis1 (Suppl Table 3). To test this notion, use for example the Hsf1 target genes that recently have been defined in Pincus et al 2018 MBoC. Is Msn2/4 activated upon Htt97Q overexpression? The targets of the general stress response pathway (Msn2/4) have been defined in Solís et al 2016 Mol Cell.

Minor concerns

1. The expression of Htt97Q does not trigger a strong activation of Hsf1, yet the data do not lend support to the claim that there is a lack of HSR activation. Instead weak HSE activation by Htt97Q is apparent in Fig 1d and in light of the suppressing effect that Htt97Q has on the reporter activity Suppl Fig 1c this may even be an underestimate. Please correct the statements of a failure to induce the HSR (line 117) and a lack of HSR (line 128).

2. For readability, please avoid one-sentence paragraphs.

3. Only some experiments are statistically analysed. To allow comparisons between two or more groups and conclusions thereof, statistical assessment of all data would be helpful. Description of statistical tests as well as fulfilled criteria for the respective analyses should be added either to the methods section or included in the respective figure legends.

4. As a courtesy to readers with colour-blindness/deficiency, please avoid red-green combinations

in micrographs (e.g. Fig 1b). Rather, use magenta-green or equivalent LUTs.

5. Fig 1b: Please briefly state in the text, why a nuclear marker was used (GFP-NLS).

6. Fig 1c and following: Please add the units (Miller Units) to the y-axis.

7. Fig 3b: This figure would be easier to understand, if the pre-bleach signal would be shown in the graph as well (as in Suppl. Fig. 3b).

8. Fig 5c: Please include protein level analysis of Sis1 to rule out that co-expression of Ssa1 compromises expression levels of Sis1.

9. Fig 5g: Please add the pre-bleaching intensity to the graph.

10. Suppl. Fig 3b: Please explain why Htt20Q does not bleach at all?

11. Suppl. Fig 4: Even though the concept of the filled/empty circles in different colours are useable, the figure would benefit if this information was conceivable without reading the figure legend.

12. Suppl. Fig 4c: Please add the respective brightfield/DIC channels to these micrographs for better visualization. Alternatively, draw the cell borders (e.g. with a white dotted line). It is hard to appreciate the number of the presented cells and the relative size of the aggregates.

13. Line 111: This statement (and others below) needs a statistical analysis (as commented above).

14. Line 153: To be able to comment on the enhancement of the deltaP variant, a wildtype Htt97Q would be needed as control in this experiment.

15. Line 162: The representative micrographs shown in Suppl. Fig 2e suggest reduced GFP intensity in Sis1-overexpressing cells. This statement would benefit from a general GFP intensity quantification.

16. Line 167: Please exchange Fig S2f by Supplementary Fig. 2f

17. Line 235: Even though there is a significant difference between wildtype and rnq1 deletion cells, the knockout shows a heatshock response, please rephrase to make this clearer.

18. Line 252: Even though the HSR is induced, the increase is about 1.5 fold, thus experiments shown before (with longer expression time) resulted in a way higher induction. Please briefly comment on this in the main text.

19. Line 262: The statement how Htt97Q is turned off would fit better above, when describing Sis1 shut-off.

20. Line 313 and Fig. 5d: The micrographs presented in Fig. 5d rather suggest a change from one dense aggregate per cell to multiple dots upon Ssa1 overexpression, similar to the Sis1-AAA mutant.

21. Line 324: Please change to Supplementary Fig. 5e

22. Line 603: Please change to "2% glucose, 2% raffinose or 2% galactose"

23. Line 604: please add: unless indicated otherwise

24. Line 605: 18 hours is most likely no longer log phase, the cells should have undergone diauxic shift. If the authors insist on the term "log/exponential" phase, a growth curve would be needed.
25. Line 615: Please add the product name of the protease inhibitor.
26. Line 636: Please move this sentence with TCA precipitation to "Preparation of cell extracts"
27. Line 642: please state from which company the benzonase was obtained.
28. Line 642: Please exchange nutation by a more commonly used term.
29. Line 663: please specify the antibodies used (organism, company, dilution,...)
30. Line 686: Please state the respective extinction and emission wavelength to detect the used fluorophors.
31. Line 737: Please add a section about the performed statistical tests (as already mentioned above)

Reviewer #2 (Remarks to the Author):

This manuscript examines the effect of polyQ aggregation on the regulation of the heat shock response, mainly using *S. cerevisiae* as a model system. Sis1, an Hsp70 J-domain protein co-chaperone, was identified in a screen of overproduction of numerous chaperones and is the focus of the report. The results of a comprehensive set of experiments demonstrate difference in character of polyQ aggregates in the presence of excess, compared to normal, levels of Sis1. This and more limited analysis in the absence of polyQ aggregates leads the authors to propose a general hypothesis of Sis1's role in regulating the activity of the heat shock transcription factor, Hsf1. Overall this is an interesting report and points to a new way of thinking about at least part of this complex regulatory system. However, analysis of the effect of Sis1 levels in the "normal" heat shock response, that is in the absence of polyQ aggregates, is incomplete.

Points.

1. Half of the model proposed in Fig 7 concerns the role of Sis1 "normal" heat shock response, that is in the absence of polyQ aggregates. Analysis of Sis1 under these circumstances should be examined and reported (indeed, this could be considered as an important control for Htt97 experiments as well). Does the distribution of Sis1 change? Is it present in (go into) observable aggregates upon a heat shock? Do the results of such analyses under levels of stress, Sis1 overexpression and Ssa1 overexpression fit the model proposed?
2. Fig 2B, in which the effect on Hsf1 activity is shown in the presence of overexpression plasmids for Sis1, Ydj1, DnaJB1 and DnaJB6, is key to the conclusions presented. However, no control for the level of expression are shown. This is a very important control, not only for Ydj1, but also to compare the relative amount of the mammalian proteins, that is B1 and B6, produced (also see comment 3).
3. Throughout the manuscript DnaJB6 is referred to as being a homologue to Sis1. This is true, as is every protein that has a J-domain, including DnaJB1 and DnaJA1/2, are homologues. However, ignoring the fact that DnaJB1 is obviously much more similar to Sis1 than is DnaJB6 is detrimental to overall clarity, and detrimental for overall clarity in the field. In some ways the DnaJB6 findings here are more intriguing because of this B1/B6 difference (and also make the controls in "2" more important).

Minor points.

4. Fig 4D. What is the overexpression level of Sis1 in these experiments. It seems to be much higher than the 5X described in other sections of the text.

5. Fig 2 E. It doesn't seem to be referred to in the text. Its point/importance is not clear.

6. Is Fig 4C mentioned in the text. What is the difference between it and Suppl. Fig 4C?

Reviewer #3 (Remarks to the Author):

The manuscript entitled "The Sis1 chaperone acts as a rheostat of the stress response to protein aggregation and elevated temperature" is a well-written and experimentally very well-executed study that is highly significant in that it begins to explain how protein aggregation under some scenarios does not induce a stress response and how the heat shock response stress-responsive signaling pathway is regulated. What is notable is that this paper supports the notion that Hsp70 titration from HSF1 is important in HSR induction (a point that may deserve a little more emphasis). The authors' experimental results support the hypothesis that Sis1-mediated activation of the HSR involves Sis1 interacting with soluble oligomers of newly synthesized polyQ expanded exon 1 in a more expanded and more soluble polyQ aggregate that permits the recruitment of Hsp70, which appears to be critical for activating the Heat Shock Response.

That almost all of the polyQ exon 1 was recovered in the soluble fraction upon Sis1 overexpression upon cell lysis, strongly suggests to me that the polyQ exon 1 aggregates are substantially remodeled by the higher level of Sis1. My protein chemistry intuition suggested that this was due to the dimeric Hsp40 functioning as a holdase chaperone with Q97-this is consistent with all of the data except "The Sis1 mutant lacking the dimerization domain (DD) preserved the ability to induce polyQ condensate formation and the stress response (Fig 5B.....)." Does deletion of the DD allow the condensates to readily dissolve upon cell lysis? Are the authors sure that Sis1 becomes a monomer when the DD is missing at the overexpression concentration? Polyvalency is the Occam's razor explanation for the gel-like increased permeability to Hsp70 while Sis1 and polyQ are much less mobile.....Is there any correlation between the 40s that allow the aggregate-mediated HSR and their quaternary structure. Of course the other explanation is that there is a higher occupancy of Sis1 per unit length of polyQ aggregates that prevents their lateral assembly that may result in insolubility.

On line 65 "fibrillar" should probably be cross-beta-sheet

Irrespective of the authors' response to our questions, this is an accept with possibly very minor revisions that I will read about when published-it doesn't make sense to hold up publication of this exciting paper for such a minor revision(s).

Response to reviewers

We thank the reviewers for their feedback and comments which were very helpful in improving the manuscript.

Reviewer #1 (Remarks to the Author):

In this manuscript Klaips et al. establish that the J-domain protein Sis1 is a limiting component for Hsf1 to induce stress-gene expression in response to aggregating polyQ. Overproducing Sis1 converts the dense, Hsp70 inaccessible PolyQ inclusions to a permeable meshwork that recruits Hsp70, thus triggering the heat shock response via liberation of Hsf1. In addition, the main findings are recapitulated using the human Sis1 homolog DnaJB6, recombinantly expressed in yeast as well as investigated in the mammalian background.

This study provides conceptual insight into how the proteostasis system is organized to detect specific misfolded species. The study suggests that Sis1 functions as a rheostat to set the sensitivity threshold of the heat shock response. Overall, the manuscript is well written and the conclusions have solid experimental support.

Major concerns

1. A major claim in the manuscript is that Sis1 functions as a rheostat of the Hsf1-driven heat shock response. The study provides convincing data in support of the notion that physiological Sis1 levels are limiting for Hsf1 activation via Htt97Q-mediated titration of Hsp70. Yet, the rheostat concept implies that physiological changes of Sis1 levels modulate Hsf1 activity. This key aspect is not developed in the manuscript, the experiments instead solely rely on genetic manipulation of Sis1 levels. This is a major limitation of the study and raises the question whether Sis1 levels really do change under different stress/growth conditions and if this translates into transcriptional rheostat. Such a demonstration would strengthen the support for the claim of a Sis1 rheostat function.

SIS1 is under regulation by HSF1 and Sis1 levels have been reported to increase during heat stress (Klaips et al., eLife 2014; Solís et al., Mol Cell 2016). However, the reviewer is correct that in the manuscript we were mostly referring to the fact that Sis1 levels are low under non-stress conditions, which we believe allows for accurate and sensitive stress sensing. We realize that the use of the word “rheostat”, particularly in the title, could be misleading in this sense. It is indeed more precise, based on the available evidence, to describe Sis1 as a limiting regulator and potentiator of the stress response. We have modified the title and text accordingly.

2. The transcriptome analysis (lines 177-203, Fig 2c-d and Suppl tables) is not clearly outlined and needs to be more explicit. Is it simply Hsf1 target genes that are induced by Sis1 overexpression in the Htt97Q condition? What other regulons are activated by Htt97Q expression (Msn2/4)? Probably volcano plots provide the most transparent method of presenting the differentially expressed genes (DEs). The DEs can be color coded in the volcano plots based on what regulons they belong to. By eyeballing, the Hsf1 regulon appears to be dominating the DEs in the comparison Htt97Q with or without overexpression Sis1 (Suppl Table 3). To test this notion, use for example the Hsf1 target genes that recently have been defined in Pincus et al 2018 MBoC. Is Msn2/4 activated upon Htt97Q overexpression? The targets of the general stress response pathway (Msn2/4) have been defined in Solís et al 2016 Mol Cell.

We have added some additional analysis and corresponding explanation to our treatment of the transcriptome data. Specifically, we have clarified that after initially comparing each of the conditions (Sis1 alone, Htt97Q alone, and Htt97Q+Sis1) to WT, we then analyze the results comparing Htt97Q alone to Htt97Q with Sis1. We have also added a volcano plot (Supplementary Fig. 2h) showing common stress regulons for the significant differentially expressed genes (DEGs) under these conditions (HSF1/MSN/UPR).

Regarding the reviewer's questions, the HSF1 regulon shows the most pronounced effects, although there is some representation from the larger MSN2/4 regulon when comparing Htt97Q vs Htt97Q+Sis1 (which we now present in the text and in Supplementary Fig. 2h). Similarly, while it is true that there are HSF1 genes represented with Htt97Q even without Sis1 overexpression, due to the large number of DEGs in this condition they represent a small fraction of the overall DEGs (the HSF1 regulon represents ~3% of all upregulated DEGs with Htt97Q alone vs. ~20% for Htt97Q+Sis1). For these reasons we have chosen to focus our discussion on the GO term analysis of all the genes to pull out the most significant categories, which we have reported for all conditions. Nonetheless, we agree that there is more information contained within this dataset outside the topic of this manuscript. The RNAseq data will be freely available upon publication.

Minor concerns

1. The expression of Htt97Q does not trigger a strong activation of Hsf1, yet the data do not lend support to the claim that there is a lack of HSR activation. Instead weak HSE activation by Htt97Q is apparent in Fig 1d and in light of the suppressing effect that Htt97Q has on the reporter activity Suppl Fig 1c this may even be an underestimate. Please correct the statements of a failure to induce the HSR (line 117) and a lack of HSR (line 128).

We have qualified these statements in the text to indicate that instead of a complete lack of HSR induction by Htt97Q, rather it is a weak or ineffective response. Additionally, we have clarified in Supplementary Fig. 1c that the suppressing effect of Htt97Q on the reporter activity does not meet significance.

2. For readability, please avoid one-sentence paragraphs.

Done

3. Only some experiments are statistically analysed. To allow comparisons between two or more groups and conclusions thereof, statistical assessment of all data would be helpful. Description of statistical tests as well as fulfilled criteria for the respective analyses should be added either to the methods section or included in the respective figure legends.

We have added additional statistical assessment to graphs where quantitative/comparative statements are made in the text (Figs. 1c,d, 3b, 4b, 5b,c, 6c, d and Supplementary Figs. 1c, 2b, c, d, j, 3, e, g, 4a, b, c, 5c, 6b, e, 7c, d, e). Also, we have added a "Statistical Methods" section to the "Methods" part of the manuscript. Additionally, in order to allow an easier examination of the distribution, individual data points have been added to all bar graphs.

4. As a courtesy to readers with colour-blindness/deficiency, please avoid red-green combinations in micrographs (e.g. Fig 1b). Rather, use magenta-green or equivalent LUTs.

Thank you for pointing this out. We have changed all micrographs to a magenta-green combination.

5. Fig 1b: Please briefly state in the text, why a nuclear marker was used (GFP-NLS).

As polyQ aggregates can also be observed in the nucleus in some systems, we wanted to ensure that the Htt97Q aggregates were occurring primarily in the cytosol. Since we performed all of our microscopy experiments in living cells, in lieu of a fixed marker such as DAPI or Hoechst staining, we used a targeted fluorescent protein. This has now been briefly explained in the text.

6. Fig 1c and following: Please add the units (Miller Units) to the y-axis.

We have added '(Miller Units)' to designate the unit of measurement to all graphs where appropriate.

7. Fig 3b: This figure would be easier to understand, if the pre-bleach signal would be shown in the graph as well (as in Suppl. Fig. 3b).

The pre-bleach signal has been added to all FRAP experiment graphs (Figs. 3e, 5g, and Supplementary Fig. 2b).

8. Fig 5c: Please include protein level analysis of Sis1 to rule out that co-expression of Ssa1 compromises expression levels of Sis1.

We have added the corresponding Sis1 expression levels for this experiment (Supplementary Fig. 5c). Ssa1 co-overexpression does not impact the level of Sis1 expression.

9. Fig 5g: Please add the pre-bleaching intensity to the graph.

The pre-bleach signal has been added.

10. Suppl. Fig 3b: Please explain why Htt20Q does not bleach at all?

Due to the small area and brief duration of the bleach, FRAP is not generally useful for measuring differences among soluble protein. The highly mobile soluble protein moves into the "bleach area" even within the 0.38s imaging frame time. Thus, it is a technique well suited for measuring differences in aggregates but not soluble material. Nonetheless, we wanted to show what a soluble protein would look like simply to illustrate that despite the fact that Ssa1 recovered quite quickly (discussed later in the manuscript, Fig. 5), it was not behaving as a soluble protein. Our results for Htt20Q are in line with previously published reports for soluble (non-expanded) Htt constructs in other systems. We have now clarified these points in the text.

11. Suppl. Fig 4: Even though the concept of the filled/empty circles in different colours are useable, the figure would benefit if this information was conceivable without reading the figure legend.

We have added additional text labels, in addition to the circles. In cases where the text was impractical, we have also added a "key" indicating the meaning of the filled/empty circles so that this set of experiments should now be better understood even without the legend.

12. Suppl. Fig 4c: Please add the respective brightfield/DIC channels to these micrographs for better visualization. Alternatively, draw the cell borders (e.g. with a white dotted line). It is hard to appreciate the number of the presented cells and the relative size of the aggregates.

We agree that it is hard to appreciate aggregate morphology at this resolution. There was also some confusion as to the difference between this data and the data presented in the main panels (old Fig. 4c, see response to Reviewer 2, point 6). To better represent this data, we have now obtained higher resolution images under all conditions (Sis1 ON/OFF, Htt97Q ON/OFF, and in combination) and include them together for comparison in the main panels (Fig. 4c), with corresponding cell borders indicated.

13. Line 111: This statement (and others below) needs a statistical analysis (as commented above).

We have added the appropriate statistical analysis (t-test).

14. Line 153: To be able to comment on the enhancement of the deltaP variant, a wildtype Htt97Q would be needed as control in this experiment.

We have added WT Htt97Q performed at the same time to this experiment as a control (Supplementary Fig. 2d).

15. Line 162: The representative micrographs shown in Suppl. Fig 2e suggest reduced GFP intensity in Sis1-overexpressing cells. This statement would benefit from a general GFP intensity quantification.

We agree there appears to be a reduced GFP intensity due to the Sis1 overexpression. However, the way this experiment is done is that the Rnq-GFP is only transiently overexpressed in order to mark the endogenous [PIN] status in our normal strains (as the transiently expressed Rnq-GFP will be incorporated into pre-existing endogenous Rnq1 aggregates). In both cases the endogenous Rnq1 is clearly visible in its aggregated form, but intensity can vary depending on the length of Rnq-GFP overexpression. We have clarified this in the text to ensure that the reader will not interpret our claim to be that Sis1 has no effect on [PIN] but rather that the prion phenotype is preserved upon Sis1 overexpression (i.e. aggregates are still present). As an additional control, we have also added examples of how [pin-] strains look under the same conditions (soluble Rnq1, Supplementary Fig. 2e).

16. Line 167: Please exchange Fig S2f by Supplementary Fig. 2f

We have corrected the text.

17. Line 235: Even though there is a significant difference between wildtype and rnk1 deletion cells, the knockout shows a heatshock response, please rephrase to make this clearer.

This experiment in Supplementary Fig. 3e was originally done in rnk1 deletion cells, which lack this prion but can still maintain other endogenous yeast prions. We think that this is the explanation for why we still saw a significant amount of aggregated PolyQ (and corresponding HSR) in these strains. For clarity, we have now repeated these experiments in true [pin-] strains. These are strains that are otherwise genetically identical from the WT strains but have been cured for all major endogenous yeast prions by treatment with GdnHCl (which inhibits Hsp104). In these [pin-] strains the results are much clearer. There is essentially no HSR even with co-expression of Htt97Q and Sis1 (Supplementary Fig. 3e). This corresponds to the soluble behavior of Htt97Q in nearly all cells (Supplementary Fig. 3d).

18. Line 252: Even though the HSR is induced, the increase is about 1.5 fold, thus experiments shown before (with longer expression time) resulted in a way higher induction. Please briefly comment on this in the main text.

While the *tet* strains are generally high expressing- indeed, over long timeframes we see the same results as with the *GPD* promoter- it is true that in the short timeframe of this experiment the overexpression of *Sis1* is less. We have now measured the *Sis1* levels at the time points used in this experiment and include this information both in the figures (Supplementary Fig. 4a) and text as a clarification as to why the response is limited at this time point.

19. Line 262: The statement how Htt97Q is turned off would fit better above, when describing *Sis1* shut-off.

Given the complexity of this experiment, we found it difficult to describe in the text the Htt97Q shutoff before it was actually used experimentally. Instead, we have added labels indicating the growth treatment used for “shutoff” or “induction” to the experimental scheme in Fig. 4a. It is worth noting that the Htt constructs used in these experiments are the same as used elsewhere in the paper (e.g. they are all expressed via a Galactose-inducible promoter, described earlier, which would be repressed upon growth in Glucose).

20. Line 313 and Fig. 5d: The micrographs presented in Fig. 5d rather suggest a change from one dense aggregate per cell to multiple dots upon *Ssa1* overexpression, similar to the *Sis1*-AAA mutant.

The example used in this set of experiments was overexposed relative to those used elsewhere. Nonetheless, the effect of *Ssa1* overexpression is still different from the AAA mutant, which shows only tiny dots and no main dense focus, and similar to WT acquired under the same settings. We do realize that it would be beneficial for readers to be able to compare images throughout the manuscript so we have now repeated this experiment using similar exposure settings used elsewhere in the manuscript for easier comparisons between the figures and now include this data, which shows the canonical “dense” aggregate, and the controls (obtained under the same settings).

21. Line 324: Please change to Supplementary Fig. 5e

The text has been corrected.

22. Line 603: Please change to “2% glucose, 2% raffinose or 2% galactose”

Our media is either supplemented with 2% glucose (for experiments without GAL induced Htt), or with 2% Raf+2% Gal (e.g. for Htt induction). We have clarified this in the text.

23. Line 604: please add: unless indicated otherwise

The text has been changed.

24. Line 605: 18 hours is most likely no longer log phase, the cells should have undergone diauxic shift. If the authors insist on the term “log/exponential” phase, a growth curve would be needed.

When we grow the cells we also dilute them so that they are actively doubling during harvest/experiment. We have clarified this in the text.

25. Line 615: Please add the product name of the protease inhibitor.

The product name has been added.

26. Line 636: Please move this sentence with TCA precipitation to "Preparation of cell extracts"

We have moved this section.

27. Line 642: please state from which company the benzonase was obtained.

We have added the company name.

28. Line 642: Please exchange nutation by a more commonly used term.

We have changed this to "rotation".

29. Line 663: please specify the antibodies used (organism, company, dilution,...)

Due to the large number of antibodies used, we have now provided this information in an additional supplementary table (Supplementary Table 7).

30. Line 686: Please state the respective extinction and emission wavelength to detect the used fluorophors.

We have added this information.

31. Line 737: Please add a section about the performed statistical tests (as already mentioned above)

We have added a section describing the statistical tests.

Reviewer #2 (Remarks to the Author):

This manuscript examines the effect of polyQ aggregation on the regulation of the heat shock response, mainly using *S. cerevisiae* as a model system. Sis1, an Hsp70 J-domain protein co-chaperone, was identified in a screen of overproduction of numerous chaperones and is the focus of the report. The results of a comprehensive set of experiments demonstrate difference in character of polyQ aggregates in the presence of excess, compared to normal, levels of Sis1. This and more limited analysis in the absence of polyQ aggregates leads the authors to propose a general hypothesis of Sis1's role in regulating the activity of the heat shock transcription factor, Hsf1. Overall this is an interesting report and points to a new way of thinking about at least part of this complex regulatory system. However, analysis of the effect of Sis1 levels in the "normal" heat shock response, that is in the absence of polyQ aggregates, is incomplete.

A more extensive analysis of the role of Sis1 in the normal heat stress response is beyond the scope of this study. However, in the revised manuscript we are interpreting the function of Sis1 in the HSR more carefully (see change in title and response to reviewer #1). Moreover, we have added new experiments (Fig. 6d, and Supplementary Fig. 7a-e) suggesting that Sis1 recruits Ssa1 to heat induced protein aggregates in a manner similar as observed with polyQ aggregates.

Points.

1. Half of the model proposed in Fig 7 concerns the role of Sis1 “normal” heat shock response, that is in the absence of polyQ aggregates. Analysis of Sis1 under these circumstances should be examined and reported (indeed, this could be considered as an important control for Htt97 experiments as well). Does the distribution of Sis1 change? Is it present in (go into) observable aggregates upon a heat shock? Do the results of such analyses under levels of stress, Sis1 overexpression and Ssa1 overexpression fit the model proposed?

We have now expanded our analysis of the role of Sis1 under conditions of the “normal” heat-induced stress response (in the absence of polyQ aggregates). We have added experiments showing the distribution of Sis1 and Ssa1 during HS (they form aggregates, Supplementary Fig. 7a). Additionally, using a model (non-amyloid) protein that forms heat-induced protein aggregates (mutant Luciferase), we now show that the principles shown for polyQ are upheld. Briefly, excess Sis1 allows for an increased incorporation of Ssa1 into these heat-denatured aggregates and a corresponding increased HSR (Fig. 6d and Supplementary Fig. 7b-e).

2. Fig 2B, in which the effect on Hsf1 activity is shown in the presence of overexpression plasmids for Sis1, Ydj1, DnaJB1 and DnaJB6, is key to the conclusions presented. However, no control for the level of expression are shown. This is a very important control, not only for Ydj1, but also to compare the relative amount of the mammalian proteins, that is B1 and B6, produced (also see comment 3).

We have now measured the relative expression levels for all the Hsp40s mentioned. Ydj1 is overexpressed at similar levels to Sis1 (new Supplementary Fig. 2b, ~5 fold). For DnaJB1 and DnaJB6, since they are not endogenous yeast proteins, we can only compare them to each other. Despite being expressed from identical constructs, we find that DnaJB6 is over-expressed more in yeast than DnaJB1 (new Supplementary Fig. 2g) which we now indicate in the text. Thus, it remains possible that DnaJB1 could replace the function of Sis1 if expressed to higher levels. We are now stating this in the text. However, the fact that all other experiments (“cloud” formation, SDD-AGE oligomer binding, etc.) work for DnaJB6 enable confidence that at least this factor can substitute for Sis1 for the process we are examining here. It is worth noting that in the mammalian experiments, DnaJB1 seems to be overexpressed more, and yet still shows less HSR potentiation than DnaJB6.

3. Throughout the manuscript DnaJB6 is referred to as being a homologue to Sis1. This is true, as is every protein that has a J-domain, including DnaJB1 and DnaJA1/2, are homologues. However, ignoring the fact that DnaJB1 is obviously much more similar to Sis1 than is DnaJB6 is detrimental to overall clarity, and detrimental for overall clarity in the field. In some ways the DnaJB6 findings here are more intriguing because of this B1/B6 difference (and also make the controls in “2” more important).

We agree that this is an interesting distinction. While DnaJB1 is clearly more homologous to Sis1 in terms of domain structure, DnaJB6 has also been shown to be functionally similar in terms of its ability to modify Htt aggregation. In addition to the experimental controls (and clarifications described above),

we have also expanded the text to better describe these differences in “structural/sequence” homology vs. “functional” homology.

Minor points.

4. Fig 4D. What is the overexpression level of Sis1 in these experiments. It seems to be much higher than the 5X described in other sections of the text.

This is the same Sis1 construct and growth conditions used in the bulk of the manuscript (GPD-Sis1). There are two reasons that the comparison looks so different. First, we are not seeing all cellular Sis1 protein in this blot. In the control without Sis1 overexpression, a substantial amount of the endogenous Sis1 is retained in the insoluble “dense” Htt97Q aggregates (Figure 3d), which do not migrate into the gel in these semi-denaturing conditions. Second, the Sis1 blot is overexposed in order to visualize the SDS-resistant oligomeric species, which is indicated in the text. This increases signal strength in the Sis1 overexpression reactions but not in the control, which lack such an increase in Sis1-associated oligomeric species.

5. Fig 2 E. It doesn't seem to be referred to in the text. Its point/importance is not clear.

Fig 2e is referred to after the discussion of the RNAseq data. The main point is to show that the transcriptional (mRNA) and translational (LacZ reporter) changes that indicate a HSR do translate to a biologically functional response in protecting cells from a lethal heat shock. We have now described this point more clearly in the text.

6. Is Fig 4C mentioned in the text. What is the difference between it and Suppl. Fig 4C?

Fig. 4c is mentioned in the text to show that transient expression of Sis1 does not remodel pre-existing polyQ aggregates, although HSR induction is observed. This is in line with a role for newly made polyQ species in inducing the HSR. This is now better described.

In contrast, the original Supplementary Figure 4c was to show the control experiment that transient shut-off of Htt97Q expression does not result in loss of polyQ aggregates. We have now combined the figures to better allow comparisons between all conditions in the new Figure 4c.

Reviewer #3 (Remarks to the Author):

The manuscript entitled “The Sis1 chaperone acts as a rheostat of the stress response to protein aggregation and elevated temperature” is a well-written and experimentally very well-executed study that is highly significant in that it begins to explain how protein aggregation under some scenarios does not induce a stress response and how the heat shock response stress-responsive signaling pathway is regulated. What is notable is that this paper supports the notion that Hsp70 titration from HSF1 is important in HSR induction (a point that may deserve a little more emphasis) The authors experimental results support the hypothesis that Sis1-mediated activation of the HSR involves Sis1 interacting with soluble oligomers of newly synthesized polyQ expanded exon 1 in a more expanded and more soluble polyQ aggregate that permits the recruitment of Hsp70, which appears to be critical for activating the Heat Shock Response.

To further emphasize the Hsp70 titration model of HSF1 activation we have added new data in Fig. 6d and Supplementary Fig. 7.

That almost all of the polyQ exon 1 was recovered in the soluble fraction upon Sis1 overexpression upon cell lysis, strongly suggests to me that the polyQ exon 1 aggregates are substantially remodeled by the higher level of Sis1. My protein chemistry intuition suggested that this was due to the dimeric Hsp40 functioning as a holdase chaperone with Q97-this is consistent with all of the data except "The Sis1 mutant lacking the dimerization domain (DD) preserved the ability to induce polyQ condensate formation and the stress response (Fig 5B.....)." Does deletion of the DD allow the condensates to readily dissolve upon cell lysis? Are the authors sure that Sis1 becomes a monomer when the DD is missing at the overexpression concentration? Polyvalency is the Occam's razor explanation for the gel-like increased permeability to Hsp70 while Sis1 and polyQ are much less mobile.....Is there any correlation between the 40s that allow the aggregate-mediated HSR and their quaternary structure. Of course the other explanation is that there is a higher occupancy of Sis1 per unit length of polyQ aggregates that prevents their lateral assembly that may result in insolubility.

The reviewer raises several interesting points that deserve further exploration by future studies. We agree that poly-valency in the interaction of Sis1 with the polyQ protein would be likely to play a role in explaining the function of Sis1 in aggregate remodeling. It does indeed seem surprising in this regard that the DD of Sis1 is dispensable, although the possibility remains that at the higher concentration Sis1 lacking the DD nevertheless forms a dimer or other kind of oligomer. One attractive possibility is that the low-complexity G/F region, which proved relevant in polyQ remodeling, may mediate such a function. It is also interesting that the mammalian DnaJB6, which could functionally replace Sis1 in yeast, lacks a DD (Supplementary Fig. 2f). Given the complexity of performing the pelleting assay in samples containing Htt97Q, we felt that it would add too much of a delay to repeat this experiment for all of the Sis1 mutants. However, based on our other analyses (HSR induction, "cloud" morphology, oligomer stabilization by SDD-AGE, etc.) it is tempting to speculate that since the DD mutant otherwise behaves as "WT", it might also allow for the formation of "dissolvable" aggregates). We have tried to give more emphasis to these aspects in the revised manuscript.

On line 65 "fibrillar" should probably be cross-beta-sheet

We have changed the text.

Irrespective of the authors response to our questions, this is an accept with possibly very minor revisions that I will read about when published-it dosen't make sense to hold up publication of this exciting paper for such a minor revision(s).

REVIEWERS' COMMENTS

Reviewer #1 (Remarks to the Author):

All my concerns have been adequately addressed.

Reviewer #2 (Remarks to the Author):

The authors have addressed by comments/questions to my satisfaction.